# Phase Synchronisation in the Kuroshio Current System

Ann Kristin Klose[1], René M. van Westen[2], and Henk A. Dijkstra[2,3]

[1]ICBM, Carl von Ossietzky University Oldenburg, Oldenburg, Germany
[2]Institute for Marine and Atmospheric research Utrecht, Department of Physics, Utrecht University, Utrecht, the Netherlands
[3]Center for Complex Systems Studies, Utrecht University, Utrecht, the Netherlands
*Correspondence to:* Henk Dijkstra <h.a.dijkstra@uu.nl>

**Abstract.** The Kuroshio Current System in the North Pacific displays path transitions on a decadal time scale. It is known that both internal variability involving barotropic and baroclinic instabilities and remote Rossby waves induced by North Pacific wind-stress anomalies are involved in these path transitions. However, the precise coupling of both processes and its consequences for the dominant decadal transition time scale are still under discussion. Here, we analyse the output of a multi-centennial long high-resolution global climate model simulation and study phase synchronisation between Pacific zonal wind-stress anomalies and Kuroshio Current System path variability. We apply the Hilbert transform technique to determine the phase and find epochs where such phase synchronisation appears. The physics of this synchronisation is shown to occur through the effect of the vertical motion of isopycnals, as induced by the propagating Rossby waves, on the instabilities of the Kuroshio Current System.

## 1 Introduction

The Kuroshio Current System (KCS) in the North Pacific plays an important role in climate through its meridional heat transport (Hu et al., 2015). Variations in Kuroshio flow patterns generate persistent sea surface temperature (SST) anomalies which influence the atmospheric midlatitude circulation and storm-track activities (Taguchi et al., 2009). Understanding the processes determining the mean path of the Kuroshio Current and its low-frequency (interannual-to-multidecadal time scale) variability are hence crucial for skillful prediction of KCS flow patterns and for determining the behaviour of the KCS under the increase of greenhouse gases and its implications for regional climate change (Wu et al., 2012).

Observations have shown that both the Kuroshio Current (KC) and the Kuroshio Extension (KE) as part of the KCS exhibit variations in their path (Taft, 1972). The KC switches between a typical Large Meander (tLM) and a Non-Large Meander (NLM) state (Kawabe, 1995) on a few years time scale. The NLM state is usually divided into the near-shore NLM (nNLM) and an off-shore NLM (oNLM) depending on the path it follows over the Izu Ridge. The KE displays transitions between a so-called elongated and a contracted state (based on the path length over a specified longitudinal range) on decadal time scales (Qiu, 2002). A contracted (elongated) state is characterised by a higher (lower) level of eddy kinetic energy (Qiu and Chen, 2010) as well as a smaller (larger) eastward surface transport, a more southerly (northerly) KE path and a weaker (stronger) southern recirculation gyre (Qiu and Chen, 2005). The relationship between KC and KE paths was investigated in Sugimoto

and Hanawa (2011). When the KC takes a tLM or nNLM path, the KE tends to prefer a contracted state and when the KC is in an oNLM state, the KE favors an elongated state.

From a theoretical point of view, the central problem is to explain the KCS path variability using the building blocks of geophysical fluid dynamics. The existence of western boundary currents in the ocean (such as the Kuroshio and also the Gulf Stream) is well explained by the linear Sverdrup-Stommel-Munk model (Sverdrup, 1947; Stommel, 1948; Munk, 1950) although properties of the western boundary layer flow in this model depend on highly idealised parameterisations of lateral and bottom friction. To explain more detailed aspects of the western boundary currents, such as their volume transport and their path variability, nonlinear stratified extensions of this model are needed.

One piece of the KCS path variability puzzle is clearly the non-stationary atmospheric forcing of the ocean flow. There is wind-stress variability over the eastern Pacific related to the Pacific Decadal Oscillation (PDO) (Mantua et al., 1997) and the North Pacific Gyre Oscillation (NPGO) (Di Lorenzo et al., 2008). Wind-stress variations cause sea surface height (SSH) anomalies through Ekman dynamics, which propagate westward as baroclinic Rossby waves (Deser et al., 1999). Altimetry data show a coincidence of the arrival of these SSH anomalies in the Kuroshio region and the transition between the KE states (Ceballos et al., 2009; Qiu and Chen, 2010). Upon the arrival of negative SSH anomalies, the KE weakens and the path shifts southward. An intensified mesoscale eddy field results from the subsequent interaction with the underlying topography, in particular the shallow topography of the Shatsky Rise, enhancing the strength of the southern recirculation gyre (Qiu and Chen, 2010). A northward migration of the KE is induced by the arrival of positive SSH anomalies (Qiu and Chen, 2010).

Another piece of the puzzle is the strong internal (or intrinsic) variability which can be generated solely through mixed barotropic/baroclinic instabilities. Studies using a hierarchy (quasi-geostrophic, shallow-water and primitive equation) of models have indicated that low-frequency path variability (both in the KC and KE) can occur under a stationary wind forcing (Schmeits and Dijkstra, 2001; Pierini, 2006; Pierini et al., 2009; Pierini, 2014; Gentile et al., 2018). Results from idealised models having a relatively high value of the lateral friction coefficient have indicated the successive (barotropic) instabilities which generate the path variability (Speich et al., 1995; Simonnet et al., 2005; Pierini et al., 2009) through oscillatory instabilities and global bifurcations. In models which also capture baroclinic instabilities, a collective interaction of the mesoscale eddies eventually leads to a so-called turbulent oscillator (Berloff et al., 2007) in which the KCS displays low-frequency variability.

Approaches on combining both forced and internal views have been made as an attempt to develop a more detailed theory explaining the path transitions of the KCS and its dominant decadal time scale in terms of the interaction between forced Rossby waves and the internal variability (including the mesoscale eddies). When a baroclinic Rossby wave model (based on the linear vorticity equation under longwave approximation) is subjected to non-stationary wind-stress forcing, no path transitions are found due to the interaction of the Rossby waves and the western boundary current (Qiu and Chen, 2010) so internal variability appears essential for KSC path variability. Using a non-linear shallow-water model, Pierini (2014) shows that the KE path length and the NPGO index to be very well correlated suggesting that the KE internal variability is excited and paced by the North Pacific wind-stress forcing.

During the last decade, also much has been learned from the analysis of simulations with high-resolution ocean general circulation models (OGCMs). The relevance of internal ocean variability in the climate system was demonstrated by comparing OGCM simulations under realistic and climatological annual cycle atmospheric forcing (Penduff et al., 2011; Sérazin et al., 2018). Taguchi et al. (2007) present a hind cast simulation analysis of the OFES model and indicate that Rossby waves generated by mid-Pacific wind stress anomalies can trigger internal variability of the KE. Recent studies of the mechanical energy balance for the KCS region in OGCMs (Sérazin et al., 2018), re-analysis (Yang et al., 2017) products (such as ECCO) and also altimetry observations (Wang et al., 2016; Ma, 2019) have shed light on the dominant energy transfer processes. Through inverse cascading processes, eddies with time scales of a few months continually loose kinetic energy to motions on longer time scales (Sérazin et al., 2018). The inverse cascading processes are hardly altered in a hindcast with full atmospheric forcing compared to the climatological simulation (Sérazin et al., 2018). Wang et al. (2016) show that the barotropic energy conversion between mean flow and eddies is the main source of low-frequency KE variability and that baroclinic processes are relatively unimportant. Yang et al. (2017) also show that the barotropic kinetic energy transfer lags the NPGO index by about 2 years.

In this paper, we contribute to the understanding of the KCS path variability refining existing results of the relation between the wind stress and KCS variability (Taguchi et al., 2007; Pierini, 2014) to reconcile the forced and internal views as sketched above. Using methods from nonlinear dynamics we here investigate a possible phase synchronisation of the KCS variability with the zonal wind-stress variability in the North Pacific. Within an idealised quasi-geostrophic model framework, phase locking regimes in the response of a subtropical gyre to periodic wind forcing were already revealed and proposed as possible cause for the observed western boundary current variability (Kiss and Frankcombe, 2016). As will be explained in more detail in Section 2 below, phase synchronisation is a fairly general mechanism where an external forcing is entrained into the behaviour of a chaotic dynamical system (Pikovsky et al., 2001). We will use here a 300-years control simulation of the Community Earth System Model (CESM, version 1.0.4) because one needs a relatively long (multi-centennial) simulation with a high-resolution global climate model to detect such a decadal time scale phase synchronisation.

We start with describing the model data and methods of analysis in Section 2. Results of the analysis of the CESM output are presented in Section 3, where we first provide results of a model-observation comparison (Section 3.1) followed by the phase synchronisation results (Section 3.2) and the physical mechanism of this synchronisation (Section 3.3). We summarise and discuss our findings in the context of previous work in Section 4.

## 2    Models and Methods

We analyse output from a 300-years control simulation of the CESM (version 1.0.4) as performed at SURFSara, the Academic Computer Centre in Amsterdam. In this control simulation (see also van Westen et al. (2018)) all the forcing fields such as $CO_2$, $CH_4$, solar forcing and aerosols are the ones from the year 2000 and repeated every year. The ocean component of CESM, the Parallel Ocean Program (POP), has a horizontal resolution of 0.1° on a curvilinear grid so that mesoscale eddies are explicitly captured. The POP has 42 non-equidistant vertical layers where the highest vertical resolution is near the surface. The

atmospheric component with 30 non-equidistant pressure levels and the land-surface component have a horizontal resolution of 0.5°.

## 2.1 Model validation

From the model output, we extracted monthly averaged fields of sea surface height (SSH), zonal and meridional wind stress, ocean horizontal velocity fields, ocean temperature and ocean salinity. For analysis these fields were transformed onto a rectangular grid with a horizontal resolution of 0.4° × 0.4°. Only the last 151 model years of the CESM output (model years 150 – 300) are considered in the analysis because the upper ocean quantities are not in equilibrium yet during the first 150 years (van Westen et al., 2018).

In Figure 1a, the standard deviation of the model SSH field in the KCS region is plotted over the model years 250 – 275 (26 years) and is compared to the standard deviation field of the monthly averaged SSH from AVISO (https://www.aviso.altimetry.fr/) over the years 1993 – 2018 in Figure 1b (also 26 years). Overall, both the pattern and amplitude compare reasonably well, but the variability south of Japan in the CESM is higher than observed either due to overestimation by the model or due to underestimation of the true variability by the AVISO data. In CESM, the KE path length is based on the 70-cm SSH contour in the region along 140°E – 160°E and its variations are shown for two specific model years in Figure 1c. The time series of the KE path length is shown in Figure 1e. The red curve is the 36-months running mean of the path length time series and clearly displays decadal time scale variability. Compared to the similarly computed AVISO path variations (Figure 1d), based on the 100-cm SSH contour, and the time series of its path length (Figure 1f) over the period 1993 – 2018, the amplitude and time scale of the variations is well simulated in CESM. The results in Figure 1 show that, although the model forcing is fixed at year 2000 values, the KE path variability and the overall levels of SSH variability are well captured and hence the model appears to be fit for purpose to study the processes controlling the path variability in the KCS.

## 2.2 Phase synchronisation analysis

Phase synchronisation in chaotic oscillators was introduced by Rosenblum et al. (1996) and is extensively described in the book of Pikovsky et al. (2001). While phase synchronisation is often defined as the locking of phases for the case of periodic self-sustained oscillators, a bounded phase difference can also occur in the case of chaotic oscillators (Osipov et al., 2003). Synchronisation of phases has been assessed using different methods (Quiroga et al., 2002; Boccaletti et al., 2001) in a wide range of applications, for example in the climate system (Maraun and Kurths, 2005; Jajcay et al., 2018; Gelbrecht et al., 2018; Paluš and Novotná, 2006; Feliks et al., 2010), in ecology (Blasius et al., 1999), and in physiology such as the cardiorespiratory system (Bartsch et al., 2007; Schäfer et al., 1999).

In order to detect epochs of synchronisation in the KCS we have to choose two time series representing the processes which are supposed to synchronise. For the KE variability, we choose the KE path length time series, as plotted in Figure 1e but then over the full 150 year period (cf. Figure 2c below). This time series was smoothed by a running mean of 36 months and was normalised by its standard deviation. The KE path length was chosen because it appeared as a very good indicator of the

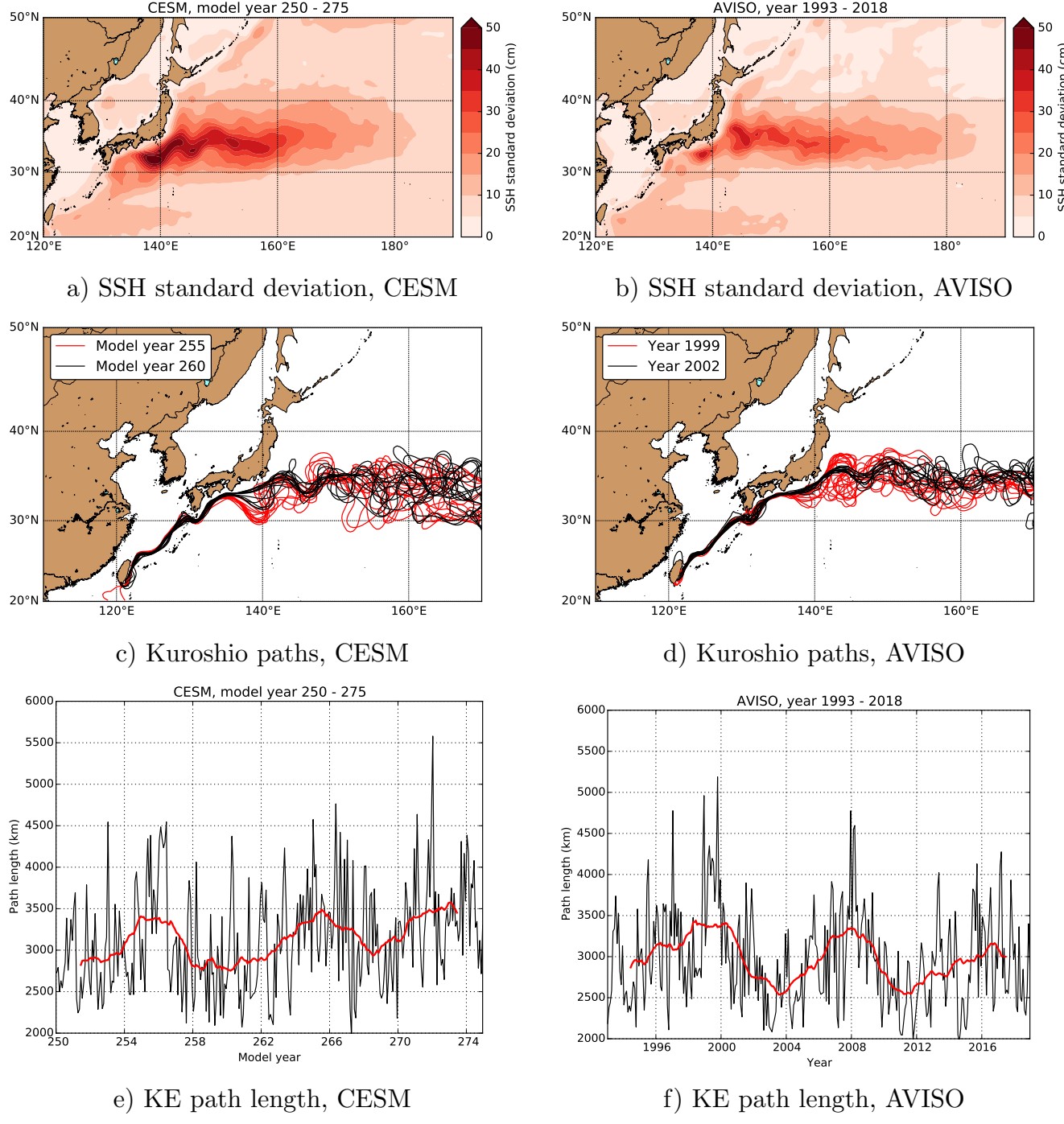

**Figure 1.** (a): Standard deviation of the SSH field over model years 250 – 275 in CESM. (b): Same for SSH field in the AVISO data over the period 1993 – 2018. (c & d): Monthly mean paths of the Kuroshio for two specific years in CESM and AVISO. (e): Path length of the KE in CESM as determined through the 70-cm SSH contour along the region $140°$E – $160°$E. (f): Path length of the KE in AVISO data, similarly computed as for CESM, but using the 100-cm SSH contour. In e) and f), the red curves are the 36-months running mean of the original time series.

different KCS states in many previous studies (Pierini et al., 2009; Qiu and Chen, 2010). The zonal wind-stress anomaly field over the North Pacific ($100°$E – $260°$E $\times$ $20°$N – $50°$N) was taken as representative for generating the Rossby waves affecting the KCS. Each time series of this field was linearly detrended, the seasonal cycle was removed and then it was normalised by its standard deviation. The resulting time series was then scaled by the area of the grid cell with respect to the largest grid cell area (which is set to 1) and are then processed by Principle Component Analysis (PCA, Preisendorfer (1988)). The time series corresponding to the first Principal Component (PC) is used for further analysis (cf. Figure 2d below). This PC time series was also smoothed by a running mean of 36 months and normalised by its standard deviation.

Motivated by the phase synchronisation studies of Paluš and Novotná (2006), Feliks et al. (2010) and Gelbrecht et al. (2018), a Singular Spectrum Analysis (SSA, Ghil et al. (2002)) is applied to the time series of the KE path length and the first PC of the zonal wind stress to extract significant oscillatory components. In the SSA, the time series is shifted up to a predefined lag $M$, i.e. the time series is delay–embedded resulting in a $M \times N$-matrix. The choice of $M$ is a trade-off between applying a wide window to obtain as much information as possible and a window size which includes many replicates of the time series properties of interest to assure statistical confidence (Ghil et al., 2002). A PCA is performed for the lag-covariance matrix of the time-delay embedded time series. A Monte Carlo SSA (Allen and Robertson, 1996; Allen and Smith, 1996) is used to test the significance of the components filtered out from each time series against red noise as null hypothesis. In the results below, confidence intervals in the Monte Carlo SSA were calculated using 2500 surrogate realisations. By reconstruction of parts of the original time series, reconstructed components (RCs) are obtained (see Ghil et al. (2002) for further mathematical details). RCs resulting from the SSA applied to the first PC of the zonal wind stress and the KE path length are used in the phase synchronisation analysis which is described in the following.

A common approach to derive the phase from a scalar time series $x(t)$ is a two-dimensional embedding by calculating the analytic signal (Gabor, 1946) using the Hilbert transform. First, the time derivative $\dot{x}(t)$ is calculated using a second order differencing scheme to centre the oscillations around the origin and to assure correct phase detection (Osipov et al., 2003). The instantaneous phase in the two dimensional embedding of the signal is then given by:

$$\phi(t) = \arctan\left(\frac{H(\dot{x}(t))}{\dot{x}(t)}\right) \tag{1}$$

where $H(\dot{x}(t))$ is the Hilbert transform of the time derivative $\dot{x}(t)$. Subsequently, the phase difference of two time series $x_i(t)$ and $x_j(t)$ is calculated as

$$\Delta\phi(t) = \phi_i(t) - \phi_j(t) \tag{2}$$

Epochs of phase synchronisation appear as plateaus in the phase difference curve. To avoid a selection of plateaus in the phase difference evolution purely based on visual inspection, we follow Gelbrecht et al. (2018) who provided a statistical test. First, sliding windows of different lengths were applied to compute the phase differences $\Delta\phi(t)$ and for each window the phase difference was mapped back to the interval $[0, 2\pi]$. The distribution of these 'mapped-back' phase differences was then calculated using a histogram with 16 bins. A peak in the distribution of these phase differences at a preferred phase difference then indicates a phase synchronisation while a uniform distribution without preferred phase difference will result from phases of variables having no relation to each other.

In the results below, 1000 iterative amplitude adjusted Fourier transform (iAAFT) surrogates (Schreiber and Schmitz, 1996) have been generated to test the significance of possible peaks in the histograms. We calculated the $k$-th surrogate phase difference as (Gelbrecht et al., 2018)

$$\Delta\phi_k^{(s)_k}(t) = \frac{1}{2}\left(\left(\phi_i(t) - \phi_j^{(s)_k}(t)\right) + \left(\phi_i^{(s)_k}(t) - \phi_j(t)\right)\right) \tag{3}$$

where the $(s)_k$ refers to the surrogates. The phase difference distribution of the actual data was then compared to the phase difference distribution of the surrogates using a Kolmogorov-Smirnov (KS) test, with a uniform distribution as a null hypothesis of no-synchronisation. A low p-value ($p < 0.0001$) indicates a rejection of the null hypothesis.

Although many other methods and measures of phase relationships have been used (Paluš et al., 2007; Vejmelka et al., 2009; Jajcay et al., 2018; Paluš, 2014a, b), we chose the approach above because the plateaus in the phase difference evolution as epochs of phase synchronisation are very intuitive compared to e.g. the Shannon entropy measure (Paluš et al., 2007) or mutual information (Jajcay et al., 2018; Paluš, 2014a, b). In addition, the approach has been applied in other geophysical problems before (Maraun and Kurths, 2005; Paluš and Novotná, 2006; Gelbrecht et al., 2018), with a preference for the analytic signal (Gabor, 1946) method to derive the instantaneous phase over alternatives, such as the wavelet transform (Paluš and Novotná, 2006). A comparison of the phase difference evolution when derived using the analytic signal and the wavelet transform showed that both methods results in the same plateaus in the phase difference evolution in terms of their localisation in time (Paluš and Novotná, 2006).

## 3 Results

We first show a brief model - observation comparison in Section 3.1 to further demonstrate that the model is fit for purpose to analyse KCS path variability. Next, the main results of the paper are provided in Section 3.2, where the phase synchronisation is studied, and in Section 3.3 where the phase synchronisation mechanism is analysed.

### 3.1 Variability

Figure 2a shows the monthly mean SSH field for January of model year 250 with the 70-cm SSH isoline in black. For the displayed model month, the Kuroshio Current meanders offshore and returns to the coast at approximately 35°N. Here the Kuroshio separates from the coast, forming the meandering eastward flowing KE into the open ocean. The first EOF of the basin wide zonal wind stress (Figure 2b) accounts for 29.6% of the total variance and resembles that of the observed PDO pattern where the explained variance is 25% (see e.g. Figure 10 in Deser et al. (2010)). Figures 2c and 2d show the KE path length along the region 140°E – 160°E as well as the first PC of the zonal wind stress over the North Pacific basin (100°E – 260°E × 20°N – 50°N).

Before determining the Fourier spectrum, each of the time series is linearly detrended and the seasonal cycle is removed. Surrogate red-noise (2000 realisations) is generated to determine the confidence levels; note that for the wind-stress the lag 1-autocorrelation is small, resulting in effectively white-noise surrogates. Figures 3a – b show the Fourier spectra for the two time

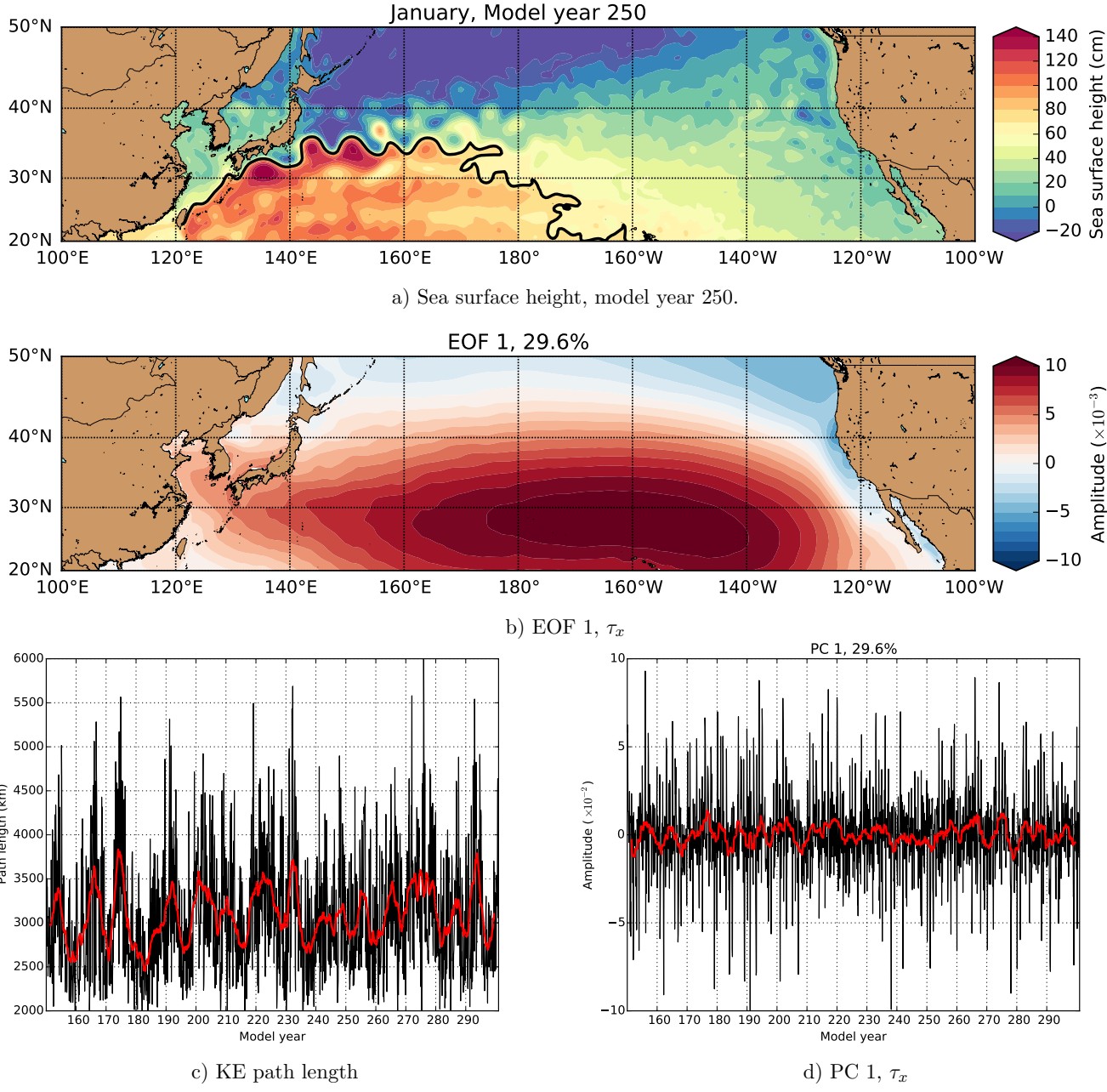

a) Sea surface height, model year 250.

b) EOF 1, $\tau_x$

c) KE path length

d) PC 1, $\tau_x$

**Figure 2.** (a): Sea surface height field in the North Pacific basin for January model year 250. The black curve is the 70 cm SSH isoline that represents the KE jet. (b): First EOF of the zonal wind stress field over the North Pacific basin ($100°$E – $260°$E × $20°$N – $50°$N), explaining 29.2 % of the total variance. (c): Time series of the KE path length along the region $140°$E – $160°$E, together with its moving average of 36 months (red curve). The path length is derived from the 70 cm SSH isoline, which can be identified in (a). (d): First PC of the zonal wind stress field over the North Pacific basin ($100°$E – $260°$E × $20°$N – $50°$N), together with its moving average of 36 months (red curve).

series and both display significant (with respect to red noise) decadal variability. Although there is a multi-decadal component of 25 year, a broad range of variability is visible in the 8 – 12 years window. The dominant period of the first PC of the zonal wind stress is 8 years. The spectra using the smoothed time series (using the moving average of 36 months, red curves in Figure 2) also resulted in dominant and significant ($> 99\%$ confidence level) periods varying between 7 – 11 years for the KE

path length and an 8-year period for the first PC of the zonal wind stress (not shown here).

When performing the SSA as described in Section 2.2 for the first PC of the zonal wind stress and the KE path length, periods of about 80 – 100 months (7 – 8 years) were found to be significant in both time series in the data basis as well as in the noise basis (Figure 3c – f). The so-called Space Time Principal Components (ST-PCs) corresponding to the SSA modes for a $M = 365$ months have been used here as a representation of the 8-year period but we have tested their robustness using

various lag-window lengths $M$ between 250 – 400 months. Values of the lag-window length $M$ outside the considered range are not expected to give meaningful results considering the trade-off described in Sec. 2.2. For most $M$ values, the ST-PCs, that are related to this 7 – 8-year period are between the seventh and ninth ST-PC for the KE path length. For the zonal wind stress, the first two ST-PCs are related to this 7 – 8-year period. These ST-PCs pairs are used to determine the RCs and the time series of these RCs are indicated by $L_M(t)$ for the KE path length and $\tau_M^x(t)$ for the zonal wind stress, where the dependence

on the lag $M$ is made explicit. These time series will be used in the phase synchronisation analysis described in the following section.

### 3.2   Phase Synchronisation

The two-dimensional embedding of the derivative of the observables with its Hilbert transform is shown in Figure 4a and 4b for $L_M(t)$ (KE path length) and $\tau_M^x(t)$ (zonal wind stress), respectively for $M = 365$ months. The trajectory in the plane

spanned by the time derivative of the observable and its Hilbert transform oscillates several times around the origin so that the definition of the phase is meaningful and the instantaneous phase can be detected reliably. Note that the multi-centennial length of the model simulation and the SSA filtering is crucial to obtain a clear phase signal. Figure 4c shows the phase difference evolution between $L_M(t)$ and $\tau_M^x(t)$ for three lag-window lengths ($M = 275$, $325$ and $365$ months). A (significant) plateau can be identified between model year 200 and 240 for all three values of $M$ indicating a phase synchronisation between the

time series.

Epochs where plateaus in the phase difference occur are confirmed by the statistical test described in Section 2.2 using a sliding window of 120 months length (Figure 4d). In this panel p-values smaller than 0.0001 for each $M$ in the KS-test allow to reject the null hypothesis related to the uniform distribution of the phase differences in the case of no synchronisation (see Section 2.2 for further explanations). The variation with $M$ shows that phase synchronisation occurs for other values of $M$ as

well (Figure 4d). Window lengths between 360 – 400 months show also a plateau from model year 190 till model year 290. Note that for some lag-window lengths (for example $M = 300 – 320$ months) we did not find an appropriate ST-PC pair which is related to the 7 – 8-year oscillation in the SSA. The value of the phase difference, and in particular of the plateaus, is different for the three displayed lag-window lengths ($M = 275$, $325$ and $365$ months) in Figure 4c. The deviation in phase difference is

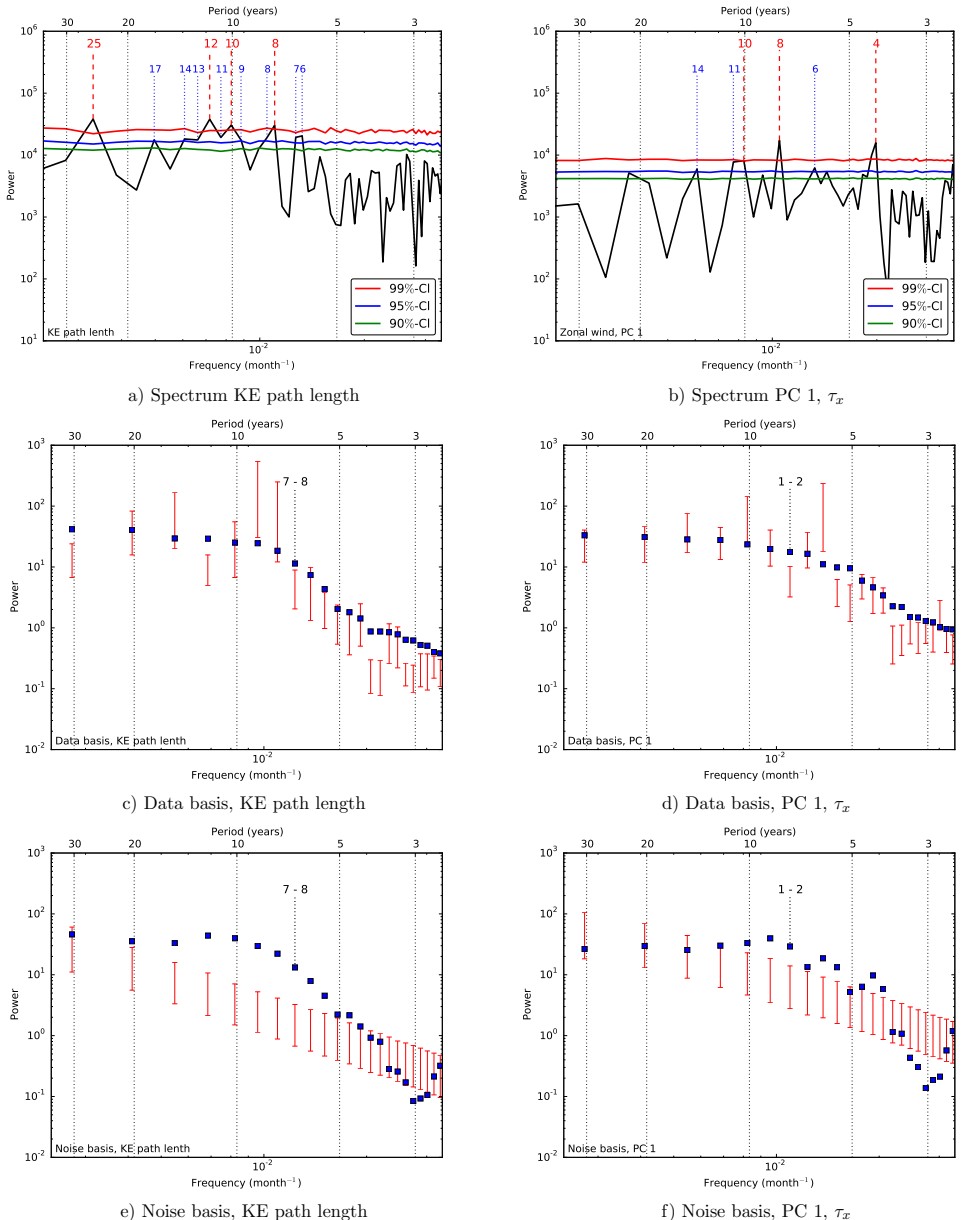

**Figure 3.** (a & b): Fourier spectrum (black curve) of the KE path length and the first PC of the zonal wind stress (obtained from the EOF analysis). The colour-coded curves are the significance levels of red-noise surrogates, where the numbers (in years) indicate significant periods. (c & d): Data basis from the Monte Carlo SSA for the KE path length and the first PC of the zonal wind stress (obtained from the EOF analysis). (e & f): Noise basis from the Monte Carlo SSA for the KE path length and the first PC of the zonal wind stress (obtained from the EOF analysis). SSA was performed with a total lag of 365 months and for the Monte Carlo SSA 2500 realisations have been used. The 95% confidence intervals which have been calculated from the surrogates are shown by the red vertical bars. A specific ST–PC resulting from the SSA is represented by the blue markers and is associated with a certain frequency. The ST–PCs related to the 8-year period are significant in both the data basis and the noise basis (Allen and Robertson, 1996) and the ST–PC pair related to this variability is indicated.

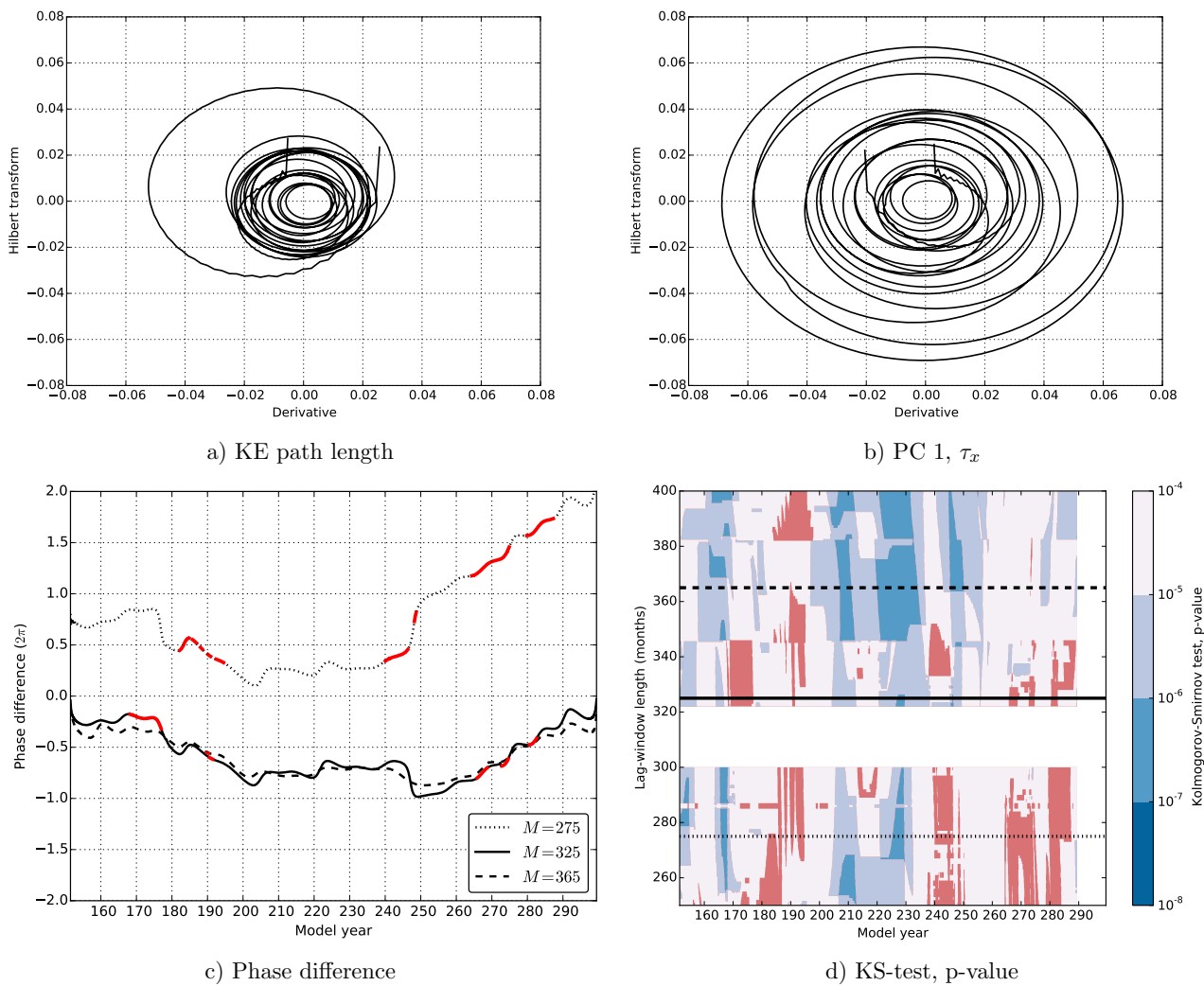

a) KE path length

b) PC 1, $\tau_x$

c) Phase difference

d) KS-test, p-value

**Figure 4.** (a & b): Embedding of the time derivative of the KE path length time series $L_M(t)$ (a) and that of the first PC of the zonal wind stress $\tau_M^x(t)$ (b) by the Hilbert transformation. (c): Phase difference between $L_M(t)$ and $\tau_M^x(t)$ for three lag-window lengths $M$. The black curves indicate starting points of the sliding window (length of 120 months) with p-value $< 0.0001$. Intervals that are not significant (p-value $\geq 0.0001$) are indicated in red. (d): Significance of the phase difference for various lag-window lengths. Regions that are not significant (p-value $\geq 0.0001$) are indicated in red. The black horizontal lines indicate the values of $M$ used in (c).

caused by different ST–PCs which arise for different lag-window lengths $M$. However, the value of the phase difference is less important than the occurrence of a constant phase difference over a given interval (i.e. plateaus).

By processing the model output data by SSA as in Gelbrecht et al. (2018), certain oscillatory components are filtered out and the phase synchronisation analysis is carried out with narrow band signals. This type of data processing introduces a risk

of observing plateaus in the phase difference evolution which are interpreted as phase synchronisation but can be technically
due to the similarity in the frequencies of the used signals (Paluš and Novotná, 2006). Here the Fourier spectra of the time
series (cf. Figure 3) show dominant periods of about 8 years for the KE path length and the zonal wind stress which are both
significant on a 95%-significance level. However, the spectra of the time series considered in the phase synchronisation analysis
(i.e. the RCs) show differences which limits the risk of detecting such spurious synchronisation. In addition, there is a good
physical picture of the coupling between the two time series which is induced by Rossby waves (cf. Section 3.3). More general
approaches to study the coupling between the time series can be found in Feliks et al. (2010) and Paluš and Novotná (2006).

In the analysis above, only the zonal component of the wind stress is considered while the wind-stress curl directly affects
the SSH field through Ekman- and Sverdrup dynamics. Therefore, we also performed a phase synchronisation analysis using
the wind stress curl over the same domain as the zonal wind stress. The first EOF of the wind-stress curl contains 7.6% of
the total variance which is considerably smaller than the variance explained by the first PC of the zonal wind stress (which is
29.6%). Conducting SSA to the first PC of the wind stress curl, a significant (95%-confidence level) ST-PCs pair was identified
with an 8-year period (against red noise surrogates). A phase synchronisation analysis between the KE path length and the
wind-stress curl revealed significant plateaus between model year 200 – 240, similar to the results for the zonal wind stress.
The similarity of the results for the zonal wind stress and the wind stress curl is expected since the meridional dependency of
the zonal wind stress is the dominant component of the wind-stress curl.
We also analysed phase synchronisation in the CESM between the NPGO index (Di Lorenzo et al., 2008) and the KE path
length using the same procedure as above. Using the KS-test, we found that there are intervals after model year 220 which
significantly differ from the uniform distribution (not shown here). However, after visual inspection (which is of course a
subjective measure), these epochs showed no clear plateau compared to those in Figure 4c. Hence, compared to the PDO
pattern, the NPGO pattern apparently does not contribute much to the excitation of Rossby waves as an essential part of the
synchronisation mechanism (see section 3.3). In addition, the dominant period of the NPGO index in the CESM output is about
19 years (95%-confidence level). Therefore it is not very likely that phase synchronisation occurs since the spectra of the KE
path length shows a dominant period of about 10 years.

Finally, it should be noted that by applying a phase synchronisation analysis and by revealing a dependence between the
phases of the observables, synchronisation cannot be deduced unconditionally. Two time series can fulfill the 'mathematical'
30  condition of phase synchronisation which is given by the boundedness of the phase difference even if the observed state arises
from other processes or effects (Rosenblum et al., 2001; Vejmelka et al., 2009). Hence, it is critical to also have a physical
picture on how this synchronisation is established.

### 3.3 Physical mechanism of phase synchronisation

Self-sustained oscillators synchronise through a coupling which allows the adjustment of the phases. Assessing the processes
responsible for the synchronisation is an important but also difficult task, especially for phase synchronisation detected from
observations. A good assessment is therefore preferably in an 'experimental setup' where parameters can be varied (Rosenblum
et al., 2001; Vejmelka et al., 2009). Formally, the performed phase synchronisation analysis does not allow to infer a direction of

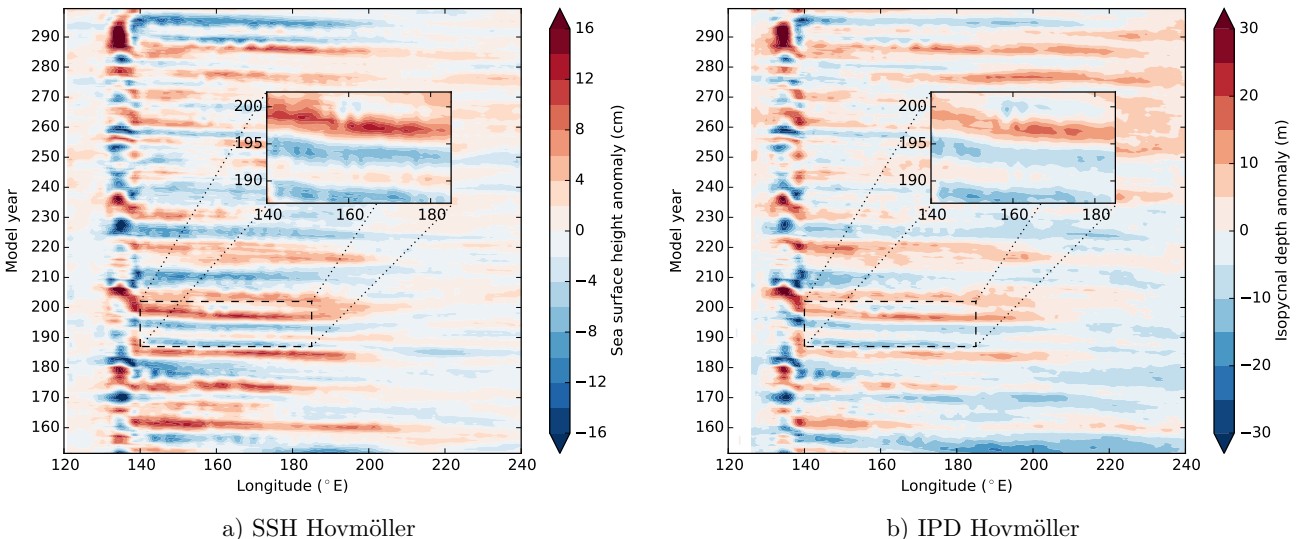

**Figure 5.** Hovmöller diagrams of a) SSH anomalies and b) IPD anomalies averaged over 28°N – 30°N. The result is smoothed by a running mean of 36 months. Positive (Negative) anomalies in the IPD means a deeper (shallower) isopycnal compared to the time-mean. The inset emphasises the (westward) propagation of the anomalies.

coupling between the proposed oscillators. Causality methods as conditional mutual information (Jajcay et al., 2018) or causal discovery algorithms (Schleussner et al., 2014; Runge et al., 2014, 2015) as well as quantifying the direction of influence (Feliks et al., 2010) between the proposed oscillators might help to reveal the direction of coupling.

However, as also mentioned above, here an adequate view on the physics of the system motivates to explore a possible coupling mechanism from the zonal wind stress to the KE path length through the propagation of SSH anomalies. We specifically also focus on the propagation of the anomalies of the depth of the 1028 kg m$^{-3}$ isopycnal, below indicated by the IPD. First, the SSH and IPD anomalies across the basin (120°E – 240°E) were averaged along latitude bands, the time-mean was subtracted and the resulting anomalies were smoothed by a running mean of 36 months. The Hovmöller diagrams of the SSH and IPD anomalies averaged over the more southern latitude band of 28°N – 30°N (Figure 5) show a westward propagation of the SSH and IPD anomalies starting from the eastern part of the basin and arriving in the KCS. The travel time of these Rossby waves from the eastern to the western side of the North Pacific basin is estimated to be roughly 10 years (Chelton and Schlax, 1996) corresponding to the observed decadal variability of the Kuroshio oscillator. We also find Rossby waves for the more northward latitude bands (tested between 30°N – 34°N) but the propagation is less clear than in Figure 5 because of the large variability in the KCS.

The Hovmöller diagrams (Figure 5) also indicate that positive (negative) SSH anomalies coincide with positive (negative) IPD anomalies, as expected from the baroclinic adjustment process. The stretching of the layer bounded by the surface and the 1028 kg m$^{-3}$ isopycnal, which is on average roughly 500 m deep in this region, is expected to lead to variability in

the horizontal velocity field through the thermal wind balance, which will affect instabilities of the KCS. To analyse how the velocity of the KCS varies due to variations in the IPD, we determined the zonally averaged zonal velocity and density between 140°E – 160°E (i.e. Kuroshio Extension region). These zonally averaged quantities are shown in Figure 6a and 6b for model years 174 –176 and 181 –183, respectively. These years represent a contracted state (model years 174 –176) and an elongated state (model years 181 –183) of the KC. For both periods of time, the upper 1000 m background flow is mainly directed eastward in the Kuroshio Extension region. Relatively warm (cold) water is found south (north) of the KC, which explains the positive meridional slope for the isopycnals. The gradient in the isopycnal slope is larger for model years 181 – 183 with respect to model years 174 –176, in particular between 33°N – 35°N. As a result of thermal wind balance, relative high zonal velocities are found between 33°N – 35°N for model years 181 – 183 with respect to model years 174 –176.

When comparing the 1028 kg m$^{-3}$ IPD between model years 174 –176 and 181 – 183, there is a clear decadal change in the background state. Variations of the 1028 kg m$^{-3}$ IPD for the complete time range (model years 150 – 300), as shown in Figure 6c, also display decadal variability. The maximum gradient in the 1028 kg m$^{-3}$ IPD slope occurs around 34°N and its value is also displayed in Figure 6d. At the latitude where the maximum gradient of the 1028 kg m$^{-3}$ IPD slope occurs (for example around 34°N for model years 181 – 183), we determined the maximum zonal velocity with depth. This maximum zonal velocity is also displayed in Figure 6d and both time series are significantly (95%-confidence level) correlated with a value of 0.96.

To determine the effects of the variations in the IPD and the background zonal velocity on the KC stability, we consider the local eddy kinetic energy (EKE) of the KC system, defined here as

$$\text{EKE} = \frac{1}{2}\left((u - \bar{u})^2 + (v - \bar{v})^2\right) \tag{4}$$

where $u$ and $v$ are the local monthly-averaged zonal and meridional velocity, respectively and $\bar{u}$ and $\bar{v}$ are their time mean. The quantities EKE and IPD are averaged over 135°E – 145°E and 30°N – 35°N, and the EKE is additionally averaged over the upper 500 m. For this spatially averaged IPD time series and for the KE path length time series (as in Figure 2c), the time-mean is subtracted. The three time series (EKE, IPD and KE path length) are smoothed with a running mean of 36 months and plotted in Figure 7a. The time series for IPD and EKE are clearly out of phase, and the IPD appears to lead the EKE by about 40 months (Figure 7b). The highest lag-correlation (0.29, significant at 95%-confidence level) between EKE and KE path length is 10 months, with EKE leading (Figure 7b). However, taking the spatial average over a larger region (for example, 140°E – 160°E × 30°N – 35°N) for the EKE resulted in higher lag-correlation coefficients with the KE path length.

We interpret the results in Figures 6 and 7 as follows. During the deepening of the IPD (time derivative positive), the isopycnal slope also decreases and, by thermal wind balance, this leads to a weaker background zonal velocity with weaker horizontal and vertical shear (Figure 6). This results in a lower intensity of mixed barotropic/baroclinic instabilities and hence relative low values of EKE (with respect to the time-mean). In more idealised configurations (Kiss and Frankcombe, 2016), one could study this in more detail by looking at the change in Floquet multipliers (associated with instabilities in the presence of periodic background forcing), but such an analysis is out of the scope of these CESM results. The nonlinear interaction of growing perturbations due to the mixed barotropic/baroclinic instabilities leads to a modification of the mean state, usually

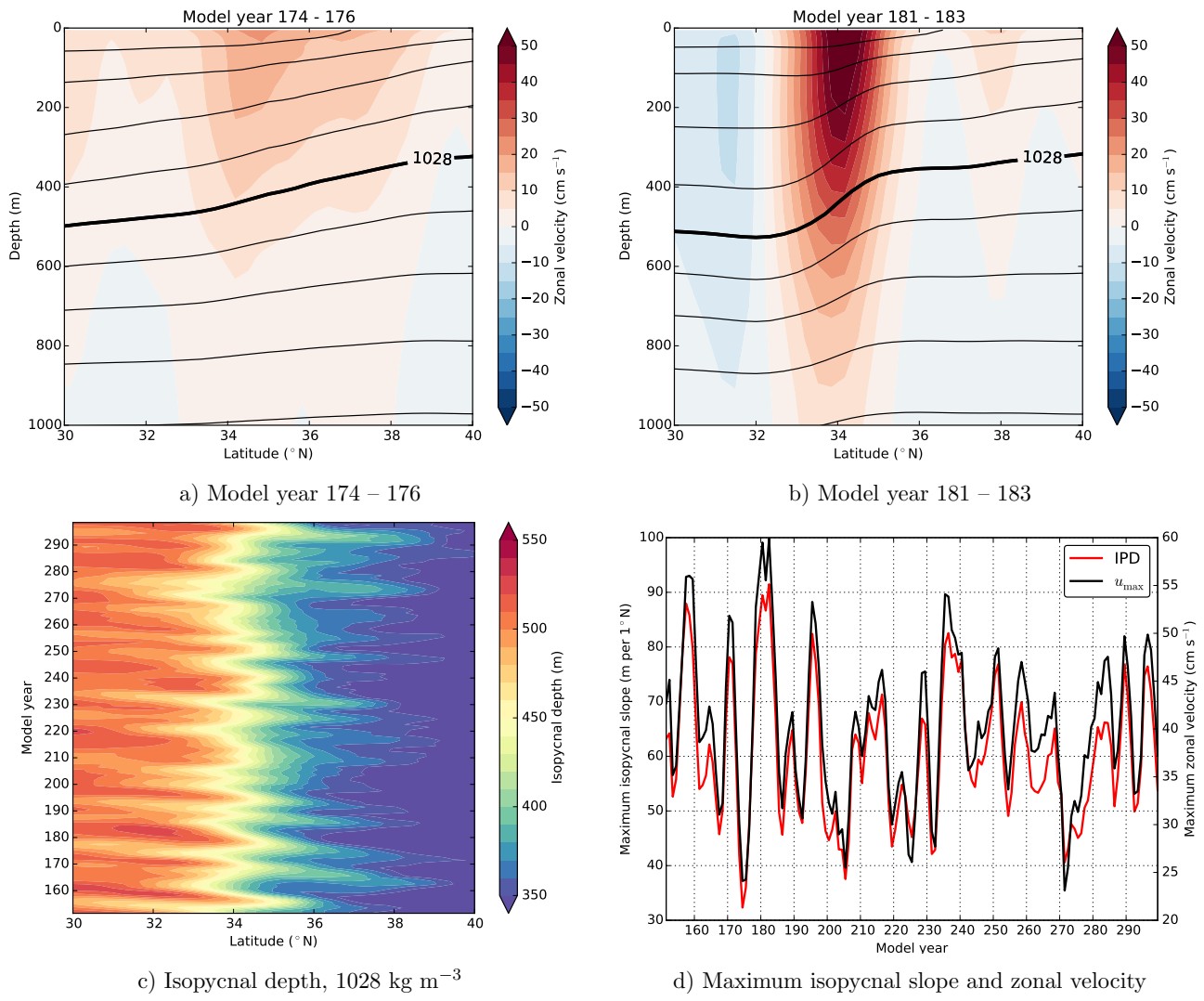

a) Model year $174 - 176$

b) Model year $181 - 183$

c) Isopycnal depth, $1028$ kg m$^{-3}$

d) Maximum isopycnal slope and zonal velocity

**Figure 6.** (a & b): The zonally averaged ($140°$E $- 160°$E) zonal velocity with depth, averaged in time (3-year average) for a) model years 174 $- 176$ and b) model years $181 - 183$. The black curves are isopycnals each spaced by 1 kg m$^{-3}$ and for reference the 1028 kg m$^{-3}$ isopycnal is indicated. (c): Hovmöller diagram for the evolution of the 1028 kg m$^{-3}$ isopycnal depth, smoothed by a running mean of 36 months. (d): Time series of the maximum slope of the 1028 kg m$^{-3}$ isopycnal and the maximum zonal velocity at the maximum slope latitude. The time series are smoothed by a running mean of 36 months.

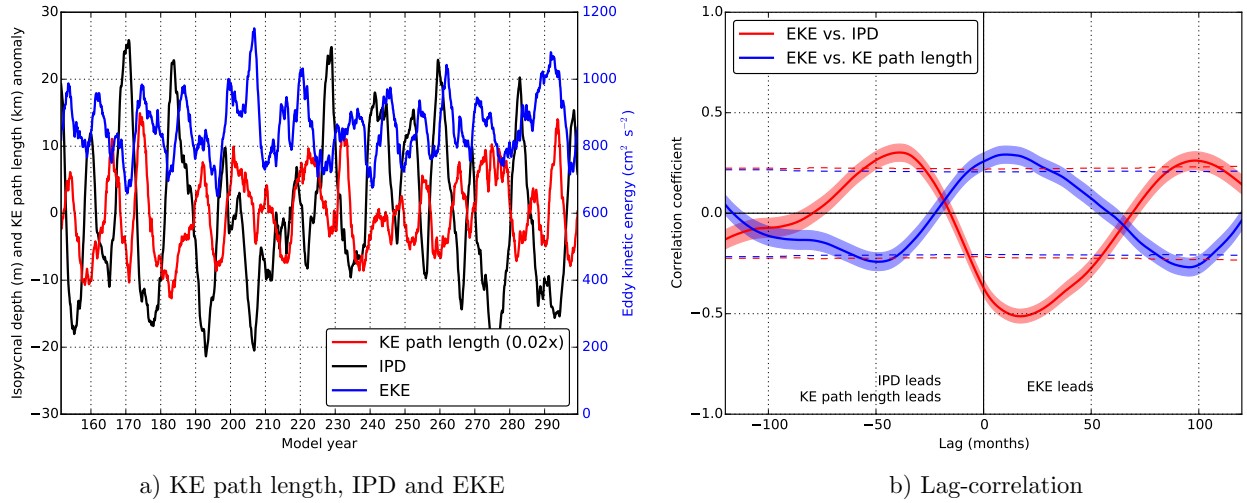

| a) KE path length, IPD and EKE | b) Lag-correlation |

**Figure 7.** (a): Time series of SSH, IPD and EKE. The time-mean is subtracted to retain anomalies for the SSH and IPD time series. All time series are smoothed by a running mean of 36 months. (b): Lag-correlation between the IPD, SSH and EKE time series. The shading indicates the 95%-confidence interval and the dashed curves indicate significant correlations (95%-confidence level.)

referred to as rectification. In the phase of decreasing EKE, rectification effects due to the instabilities become weaker and the KC evolves into a contracted state. When the IPD shallows, precisely the opposite happens as the horizontal and vertical shear increase and the degree of instability increases, leading to a higher EKE, a stronger rectification and hence to an elongated KC

10  state. The coupling of Rossby waves to the KCS system through the motion and sloping of the isopycnals and the resulting changes in the zonal velocity field is also compatible with the analysis of other results in OGCMs (Taguchi et al., 2005, 2010) indicating that the Rossby waves can 'trigger' the frontal (internal) instabilities.

## 4   Summary, Discussion and Conclusions

Output from a multi-centennial simulation with a high-resolution version of the Community Climate System Model (CESM)

15  was used to study phase synchronisation between path variability in the Kuroshio Current System (KCS) and the wind variability over the North Pacific basin. In particular, by performing an SSA, oscillatory components on decadal scales of the Kuroshio Extension (KE) path length and the first PC of the zonal wind-stress field were chosen as representative observables for the proposed oscillators. The length of the CESM simulation is crucial for determining an adequate phase of the KE path length and the zonal wind-stress variability. The high spatial resolution of the ocean model component is also needed to adequately represent the internal variability in the KCS. The phase difference evolution, the distributions of the phase difference and results from a statistical test indicate that the KE and the North Pacific zonal wind stress are episodically phase synchronised in CESM.

We interpret this phase synchronisation as an entrainment of an 'external' frequency (that of the mid-basin wind stress variability) into the variability of a chaotic system (the KCS), as described in section 5.2 of Pikovsky et al. (2001). Although there are effects of variations of the KSC on the atmospheric circulation (predominantly local) as for example described in Qiu et al. (2014), there is ample evidence that the PDO and/or NPGO variability is independent of the KCS as it is also found in low-resolution ocean/coupled models (Weijer et al., 2013). Hence, the wind-stress variability acts as a forcing on the Rossby waves which couple to the KCS system through the motion of isopycnals as shown in Section 3.3.

When this forcing is too large with respect to the intrinsic KCS variability, the period of the 'external' frequency is dominant and hence, in case of a periodic forcing, the resulting KCS would become periodic in a chaos–destroying synchronisation (as described in Pikovsky et al. (2001)), which it is clearly not. When the forcing is too small with respect to the intrinsic KCS variability, phase diffusion induced by nonlinearities in the KCS would be dominant and prevent phase synchronisation. This likely explains why only during the main interval between model year 200 to 240, the phases are synchronised. The amplitude of the zonal wind stress time series (Figure 2d) is relatively large before the synchronisation interval (model years 200 – 240) and when its amplitude drops again (model years 240 – 250), phase synchronisation disappears. Hence, the variation in the forcing sets the epochs of the phase synchronisation. Of course, there also exist (significant) plateaus in the phase difference evolution during other time intervals. However, they are not robust across the $M$ values so that the existence of phase synchronisation cannot be clearly demonstrated. Compared to the coupled CESM simulation, IPD anomalies have a smaller magnitude in an ocean only (the POP model) simulation (not shown here), which was forced by a seasonal cycle (for the simulation details, see Weijer et al. (2012)). In addition, they do not show a clear decadal variability, i.e. IPD anomalies in the climatological and coupled simulation are not comparable.

We think that the phase synchronisation as investigated with methods from nonlinear dynamics, together with its mechanism as described in Section 3.3, provides a further step to connect the purely Rossby wave 'forced' view of KCS path transitions, as originally advocated (Qiu and Chen, 2005, 2010) and the pure internal variability view (Pierini et al., 2009; Berloff et al., 2007). In the model world of CESM, the Rossby waves are clearly required to obtain a very pronounced decadal variability. This view is supported by (not shown) results we obtained with the OGCM-only (the POP model) which was forced by a seasonal cycle. While transitions between the different KE states occur regularly in the CESM output, they develop more randomly in the POP simulation because there can be no phase synchronisation.

Indications that some periods of observed variability show a much clearer decadal variability than others are found in the analysis of AVISO data (Gentile et al., 2018). However, one cannot detect the phase synchronisation from observations yet as the AVISO time series of SSH is too short. Analysis of a 26-year interval (out of the 150 year total interval) of CESM output indicated that (with a period of variability of about 8 years), it is not possible to detect such a phase synchronisation. Unfortunately, we will have to accept that we will have to wait for a few (about 5 – 6) decades before this phenomenon can be demonstrated from accurate observational data.

*Acknowledgements.* The authors thank Michael Kliphuis (IMAU, UU) who performed the CESM simulations and both anonymous reviewers for their excellent suggestions and comments, which improved the manuscript substantially. The computations were done on the Cartesius at

SURFsara in Amsterdam. Use of the Cartesius computing facilities was sponsored by the Netherlands Organization for Scientific Research (NWO) under the project 15502.

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
