# Peer review of "Phase Synchronisation in the Kuroshio Current System"

_Ocean Science, 2019_

## Referee Comment (RC1) · Anonymous Referee #1 · 10 Sep 2019

**A. GENERAL COMMENTS**

The authors analyse the outputs of a high-resolution global coupled ocean-atmosphere simulation, in order to detect and study the intermittent synchronization of the Kuroshio Current System (KCS) low-frequency (LF: interannual to multidecadal) variability with the atmospheric variability via Rossby Waves (RW). The authors aim to identify the epochs when the atmospheric and KCS variabilities are synchronized, and to propose a physical mechanisms involved in this synchronisation.

This scientific question is relevant to OS and is important since two alternative views of the KCS variability exist but still need to be reconciled: the well-known LF intrinsic variability of the KCS, and the well-known pacing of the KCS by the wind variability via westward-propagating long RW across the Pacific.

[Figure]

The statistical method used by the authors to detect synchronisation epochs is inspired from dynamical systems theories (coupling between chaotic dynamical systems), and can be applied to this large model simulation more easily than other approaches (e.g. pullback attractor, Pierini, 2018). This methodology is thus innovative and interesting (although somewhat complex), and its application is original in the context of "realistic" ocean model studies. The description of the method is available in quite theoretical papers though, and could a bit clarified in this paper for physical oceanographers.

The paper is very well written; the title, abstract and structure of the paper are well chosen. Some results are interesting and original, and the detection/analysis of synchronisation is globally convincing (although additional information is needed to strengthen the conclusions, see specific comments below).

It is clear that adding a physical interpretation to the statistical analysis would be valuable. However, the physical interpretation of the synchronisation mechanisms appears much less convincing to me. Clarifying the interpretation would require additional diagnostics, and more physically-based arguments. This remark is developed in the specific comments below (see comments starting with an "*").

B. SPECIFIC COMMENTS

1) P2 line 21: Suggestion: replace "Results from models. . ." by "Results from idealized models. . ." or "Results from barotropic models. . ."

2) P2 line 28: QG and SW models are generally nonlinear. Suggestion (if adequate): "When linearized QG or SW models. . .". If not adequate, please clarify.

3) P3 line 2: I suggest to replace ". . .can induce" with ". . .can trigger", as expressed by Taguchi et al (2007) in their abstract; I think their wording is more accurate.

4) P3 line 6: Sérazin et al (2018) also show that the temporal inverse cascade is very similar in the presence or absence of interannual forcing. This might be relevant to recall for the present study.

[Figure]

5) P4 line 4: how do SSH std maps depend on the chosen 26-year period?

6) P4 line 6: it is not sure that the model overestimates the true SSH variability; it is indeed likely that the AVISO interpolated product also underestimates the true variability.

7) * P4 line 26: It is not obvious to me why the authors chose to relate the KCS variability to the wind stress field and not to the wind stress curl field, whose direct impact on SSH (and IPD) fields through Ekman and Sverdrup dynamics is well known. The low-freq variability of the wind stress curl impacts the open ocean dynamics (and Rossby waves) in a way that is more stranghtforward than that of the zonal wind stress itself. Please justify this choice. It may be interesting to see if the link between the forcing and the ocean variability would be clearer with this alternative choice. If yes, this could be beneficial for the dynamical interpretation (see my other *remarks below).

8) P4 line 29: what is the purpose of scaling the timeseries by the grid cell areas?

9) P6 lines 7-12: this list of other methods is interesting, but makes the reader wonder why the authors chose the anylitic signal method instead of these. Could you please clarify this choice?

10) P7 line 5: how representative is a mode accounting for 29.6% of the explained variance? Is it comparable to other studies?

11) P7 lines 16-30. This part is difficult to follow for the non-specialist (that I am). I would be useful to give more explanations, about e.g. the nature and expected impact of M, the nature of the "ST-PCs", etc. Also, Figure 3's caption mentions "the first PC" and "a specific PC": are these different quantities? Please clarify these points.

12) P10 line 1: suggestion if more accurate: "...oscillates several times"

13) P10 line 4: how and why are these 3 values of M chosen? What does the detection of synchronisation for these specific 3 lag-window lengths imply physically (in terms of timescales or lags, for instance)? What is happening for other M values?

14) P10 line 4: could you please give an interpretation of the phase differences reached on the 3 "plateaus" (-0.75 and 0.25)? why do they differ among the 3 values of M, and what do these differences mean? How small should oscillations be around a given phase difference for an evolution to be labelled as a "plateau"?

15) P10 line 9: it would be interesting to show the surrogate and actual "distributions" during synchronisation.

16) P10 lines 13-18: Could you please clarify why finding dominant periods which are similar in both Fourier spectra limits the "risk" to spuriously detect a synchronisation? Line 16 indeed suggests that finding similar frequencies in both time series might expose to this risk.

17) * P10 lines 21-26: what is the implication of not finding synchronisation between the NPGO index and the KE path length? Doesn't it mean that the atmospheric pacing of intrinsic KCS variability is questionable in a way?

18) P12 lines 16-17: suggestion: "with positive (negative) IPD anomalies, as expected from the baroclinic adjustment process."

19) * P13 line 1: I do not understand the suggested dynamics, please clarify: through which simple process can the "stretching of the layer" "lead to" horizontal velocity fluctuations? Which component of the velocity? Is this a potential vorticity argument (which may lead to meridional velocity anomalies)? Alternately or in addition, would it be useful to try to relate the meridional gradient of SSH or IPD (which may lead to changes in upper zonal velocities, i.e. the strength of the zonal jet)? Trying to relate dynamically-connected fields could help interpret the link between fluctuations in SSH or IPD, the jet intensity (and perhaps the EKE).

20) P13 line 13: if "it is not entirely clear which time series leads or lags", is it OK to state a few lines above that IPD leads EKE which leads the KE length? This is a bit confusing.

21) * Last paragraph in section 3: this part is rather allusive and not solidly grounded physically (although I acknowledge that answering the following questions may not be easy given the chosen indices; see remark 19 above). - Through which process, or according to which dynamical balance, does the "deepening of the IPD [...] leads to a weaker flow" (line 14)? - Why only barotropic instability is mentioned, although isopycnal depth modulations more likely reveal changes in the 1st-mode baroclinic flow (and potentially in vertical shear and baroclinic instability strength)? It could be interesting to look at the meridional slope of the isopycnal, which relates physically to the zonal flow and to the available potential energy reservoir (hence potentially to EKE changes through baroclinic instability). - In line 16: what does "optimal" lag-correlation mean, and why is it expected to occur "at about 1/2 of the mean oscillation period"? - Finally (line 1 next page), it is not clear to me what "This view" includes precisely, given the large number of suggested links mentioned and the numerous statements made in this paragraph. Please clarify these various issues, and the dynamical arguments.

22) P14 lines 19-20: "too large/small" with respect to what?

23) P14 line 20: I think that in this context ("strong" forcing) the KCS variability would be "periodic" only if the forcing is periodic as well. If yes, please clarify.

24) P14 line 21: "during a few intervals": only one interval of synchronisation is presented in the paper. When to the other intervals occur?

25) P14 lines 22-23: I do not see a drop in the IPD index in years 240-250 (it seems to remain at about -10 on the plot, i.e. a rather moderate negative value compared to the rest of the time series). Also, are such IPD anomalies restricted to the coupled simulation (with interannual atmospheric variability), or are comparable anomalies also present in the climatological simulation (with no interannual forcing)? If comparable IPD changes occur in the climatological simulation, then one may not conclude that "the epochs of the phase synchronisation" are set by the "variation in the forcing", but by IPD changes, whether they be atmospherically driven or spontaneously produced

by oceanic non-linearities (also see next remark).

25) P14 lines 28-30: It is not fully clear what the seasonally-forced run teaches us regarding the role of RW in the decadal variability. Could you please clarify? Note that Penduff et al (2011; bottom left paragraph in their page 5663) mention that in a 1/4° climatological simulation, such large-scale RW (and LF Kuroshio variability) do exist at these latitudes despite the absence of any interannual forcing.

C. TECHNICAL COMMENTS

1) P3 line 13: 300-year control

2) P7 line 9: "lag 1-autocorrelation" is ambiguous. I suggest "1-month lag autocorrelation", if adequate.

3) Figs 4a,b: It would be nice to mark the starting point of both looping timeseries.

4) Fig 4c vertical axis label: does "(2pi)" means that the phase difference was scales by pi? By 2.pi?

5) Fig 6: only 2 panels are shown but the caption mentions 4 of them. Also, I suggest to stretch panel 1 horizontally across the available page width to increase readability.

6) P14 line 30: are the authors referring to Weijer et al (2014) instead of 2013?

---

## Referee Comment (RC2) · Anonymous Referee #2 · 15 Oct 2019

The KE low-frequency variability is known to be synchronized with the large scale wind forcing through the action of westward-travelling baroclinic Rossby waves generated in the central and eastern North Pacific Ocean. However, Rossby waves have broad spatial scales and are linear whereas the KE variability has frontal scales and is highly nonlinear and intrinsic, so, through what mechanisms can these two very different oceanic dynamical features be synchronized? Several modelling studies have been devoted to investigating this problem.

In this manuscript, the authors use a 300-years control simulation of the Community Earth System Model to study such decadal time scale phase synchronisation. The manuscript begins (sect. 2.1) with an interesting model validation by using the monthly averaged SSH from AVISO over the years 1993-2018. In sect. 2.2, after a brief review

of the mathematical methods available to identify synchronization, a detailed description of the one adopted, based on a Hilbert transform technique is provided. The analysis of the variability of both datasets and the corresponding phase synchronization are discussed in sects. 3.1-2. Finally, the physical mechanism of phase synchronization is discussed in sect. 3.3. Further analyses are reported in section 4.

The analysis is novel and sheds some new light on the problem of the KCS timing. The adopted mathematical technique is very interesting and could be applied to other problems in physical oceanography. The manuscript is fairly well written (although this can be improved, see some minor comments below). Therefore, in my opinion the manuscript could be considered for publication after a revision that should take the following comments into account.

**MAJOR COMMENT**

The open ocean generation of baroclinic Rossby waves requires two fundamental elements: time-dependence of the atmospheric forcing and perturbations of the ocean thermocline. The latter are produced by the vertical Ekman pumping associated with the horizontal divergence of the Ekman transport, which, in turn, is proportional to the vertical component of the curl of the surface wind stress So I wonder why the authors have chosen the zonal wind stress tau1 instead of curl(tau) as the representative observable for the atmosphere. Their choice is even more surprising because, to investigate the synchronization they analysed "the vertical motion of the isopycnals" induced by Rossby waves in the KE region; this is absolutely correct but, then, why ignoring the same effect in the open ocean where the waves are generated?

In my opinion the best thing would be to redo all the calculations starting from curl(tau) instead of tau1. Alternatively, I recommend to calculate the first PC of the vertical component of curl(tau), compare it with Fig. 2d and show that the two time series are, in their turn, fairly well synchronized. This is possible; in fact, if on the one hand, in the extreme -unrealistic- case of latitude-independent zonal wind stress no wave genera-
tion would occur, on the other hand, some degree of synchronization between the first PC of tau1 and that of curl(tau) can be expected. In any case I strongly recommend the authors to address this important issue.

**MINOR COMMENTS**

The mathematical method used in the analysis is relatively complex and requires the implementation of several steps, so it is not easy to follow the description. Besides, the listing of the technical steps does not end in sect. 2.2. but continues in sect. 3.1 on the variability. In particular, it is difficult to follow lines 8-30 of page 7. Therefore I suggest to compact the discussion of all the technical details in sect. 2.2 and to add a flowchart summarizing the mathematical technique. This would be very helpful, especially because I believe that the Hilbert method does not belong to the typical background of oceanographers.

p. 1, I. 22-24: the description of the elongated KE mode corresponds to that of the contracted mode, and vice versa, please exchange.

p. 2, l. 15-16: I do not think a critical effect of topography has been recognized to determine "An intensified mesoscale eddy field". Please provide references.

p. 3, I. 9-11: I do not fully agree with the sentence: "... focussing on what we think is a missing piece to reconcile the forced and internal views as sketched above: a phase synchronisation of the KCS variability with the zonal wind-stress variability in the North Pacific". Such phase synchronization has indeed been investigated in several modelling studies, several of them also quoted by the authors. I would therefore suggest to use a more moderate sentence, and similarly elsewhere.

p. 4, l. 8: define the path length parameter in the text (now it is defined only in the caption to Fig. 1) and discuss why it has been chosen among the many available parameters applied to the KE variability in previous studies.

p. 12, l. 12-14: "The travel time of these Rossby waves from the eastern to the western

OSD
side of the North Pacific basin is estimated to be roughly 10 years corresponding to the observed decadal variability of the Kuroshio oscillator." First of all the travel time of 10 years is much large than that observed by several authors (which is of 2-3 years). Secondly, the decadal variability of the KCS may be linked to that of the atmosphere but not to the travel time of the waves. Please clarify the first point and reformulate the sentence.

Section 4 includes, summary, discussion and conclusions. I think it would be much better to move the discussion about some crucial points (e.g., p. 14, I. 19-24, 28-31) in section 3.3.

p. 13, l. 16: his -> is

---

## Author Comment (AC1) · 28 Nov 2019

**Response to Anonymous Referee RC 1**

We thank the referee for the careful reading and the useful comments and will adapt the manuscript accordingly. Below is a point by point reply with the referee's comments in bold font, our reply in italic font and the changes in manuscript in normal font.

Comment of the referee:
**1) P2 line 21: Suggestion: replace "Results from models: : :" by "Results from idealized models: : :" or "Results from barotropic models: : :"**

Author's reply:
*Agreed.*

Changes in text:
Text will be changed to "Results from idealized models … ."

Comment of the referee:
**2) P2 line 28: QG and SW models are generally nonlinear. Suggestion (if adequate): "When linearized QG or SW models: : :". If not adequate, please clarify.**

Author's reply:
*Qiu & Chen (2005, 2010) use a wind-forced baroclinic Rossby wave model (based on the linear vorticity equation under longwave approximation) and observed wind stress curl data to hindcast the SSH anomalies and to qualitatively confirm the observed connection between the PDO and the SSH anomalies.*

Changes in text:
The statement of QG and SW models will be corrected in the text to better reflect the work done in Qiu & Chen (2010).

Comment of the referee:
**3) P3 line 2: I suggest to replace ": : :can induce" with ": : :can trigger", as expressed by Taguchi et al (2007) in their abstract; I think their wording is more accurate.**

Author's reply:
*Agreed.*

Changes in text:
Text will be changed accordingly.

Comment of the referee:
**4) P3 line 6: Sérazin et al (2018) also show that the temporal inverse cascade is very similar in the presence or absence of interannual forcing. This might be relevant to recall for the present study.**

Author's reply:
*Agreed.*

Changes in text:
We will add this aspect of their results.

Comment of the referee:
**5) P4 line 4: how do SSH std maps depend on the chosen 26-year period?**

Author's reply:
*SSH sdt maps were tested for various 26-years periods and did not depend on the chosen 26-year period except for some minor differences.*

Changes in text:
No changes.

Comment of the referee:
**6) P4 line 6: it is not sure that the model overestimates the true SSH variability; it is indeed likely that the AVISO interpolated product also underestimates the true variability.**

Author's reply:
*Agreed.*

Changes in text:
Text will be changed to "Overall, both the pattern and amplitude compare reasonably well, but the variability south of Japan in the model is higher than observed either due to overestimation by the model or due to underestimation of the true variability by AVISO.

Comment of the referee:
**7) * P4 line 26: It is not obvious to me why the authors chose to relate the KCS variability to the wind stress field and not to the wind stress curl field, whose direct impact on SSH (and IPD) fields through Ekman and Sverdrup dynamics is well known. The lowfreq variability of the wind stress curl impacts the open ocean dynamics (and Rossby waves) in a way that is more stranghtforward than that of the zonal wind stress itself. Please justify this choice. It may be interesting to see if the link between the forcing and the ocean variability would be clearer with this alternative choice. If yes, this could be beneficial for the dynamical interpretation (see my other *remarks below).**

Author's reply:
*We have redone the phase synchronization analysis using the wind stress curl. In particular, we calculated the first PC of the wind stress curl in a PCA which is shown in Fig. 1d. After smoothing, it is similar to the one obtained for the zonal wind stress. The EOF pattern of the wind stress curl (Fig. 1b) is different to the one obtained for the zonal wind stress and the variance for the first EOF of the wind stress curl (7.6%) is much smaller compared to the zonal wind stress (29.6%). The spectrum of the first PC of the wind stress curl $\nabla \times \tau_M(t)$ (Fig. 2b) still shows a dominant period of 8 years. Afterwards, a SSA was applied to the first PC of the wind stress curl. The same dominant period of 8 years was found. The corresponding ST-PCs pair consist of ST-PC 3 and 4 (for the zonal wind stress the ST-PC pair associated with a period of 8 years this was 1-2) and is indicated in Fig. 2d & f. For the ST-PCs pair 3-4 the reconstructed component $\nabla \times \tau_M(t)$ was calculated. Note that the results of the SSA are given for a lag of M = 325 months in the case of the wind stress curl while for the zonal wind stress results were given for a lag of M = 375 months.*
*As a next step, a two-dimensional embedding of $\nabla \times \tau_M(t)$ with its Hilbert transform was done. Oscillations of the trajectory in the plane spanned by the (time derivative) of $\nabla \times \tau_M(t)$ and its Hilbert transform around the origin can be observed (Fig. 3b).*
*Results for the phase synchronization analysis are shown in Fig. 3c & d for the $\nabla \times \tau_M(t)$ and the $L_M(t)$ (i.e. the time series obtained after performing a SSA for the KE path length). Fig. 3c shows the phase difference evolution between $L_M(t)$ and $\nabla \times \tau_M(t)$ for three lag-window lengths (M = 275, 325 and 365 months). As for the zonal wind stress, a (significant) plateau can be identified between model year 200 and 240 for all three values of M indicating a phase synchronization. Epochs with plateaus in the phase difference evolution are confirmed by the statistical test using a sliding window of 120 months length (Fig. 3d). The phase synchronization between model year 200 and 240 occurs*

*for various values of lags M. For lags M > 340 Fig. 3d also reveals a plateau for model years 250 to 290.*

*All in all, using the wind stress curl in the phase synchronization analysis resulted in very similar results as for the zonal wind stress with some minor changes such as the precise ST-PCs pair.*

[Figure]

a) Sea surface height, model year 250.

b) EOF 1, $\nabla \times \tau$

c) KE path length

d) PC 1, $\nabla \times \tau$

*Figure 1: Sea surface height field in the North Pacific basin for January model year 250. The black curve is the 70 cm SSH isoline that represents the KE jet. (b): First EOF of the wind stress curl over the North Pacific basin, explaining 7.6 % of the total variance. (c): Time series of the KE path length along the region 140°E – 60°E, together with its moving average of 36 months (red curve). (d): First PC of the wind stress curl over the North Pacific basin, together with its moving average of 36 months (red curve).*

[Figure]

Figure 2: (a & b): Fourier spectrum (black curve) of the KE path length and the first PC of the wind stress curl. The colour-coded curves are the significance levels of red noise surrogates, where the number (in years) indicate significant periods. (c & d): Data basis from the Monte Carlo SSA for the KE path length and the first PC of the wind stress curl. (e & f): Noise basis from the Monte Carlo SSA for the KE path length and the first PC of the wind stress curl. SSA was performed with a total lag of 325 months and for the Monte Carlo SSA 2500 realizations have been used. The 95% confidence intervals which have been calculated from the surrogates are shown by the red vertical bars. A specific ST-PC is represented by the blue markers and is associated with a certain frequency. The ST-PCs related to the 8-year period are significant in both the data basis and the noise basis (Allen & Robertson, 1996) and the ST-PC pair related to this variability is indicated.

[Figure]

c) Phase difference                    d) KS-test, p-value

Figure 3: (a & b): Embedding of the time derivative of the KE path length time series $L_M(t)$ (a) and that of the first PC of the wind stress curl (b) by the Hilbert transformation. (c): Phase difference between $L_M(t)$ and $\nabla \times \tau_M(t)$ for three lag-window lengths M. The black curves indicate starting points of the sliding window (length of 120 months) with p-value < 0.0001. Intervals that are not significant (p-value ≥ 0.0001) are indicated in red. The black horizontal lines indicate the values of M used in (c).

Changes in text:
A remark on the similarity of the phase synchronization analysis for the zonal wind stress and the wind stress curl will be added.

Comment of the referee:
**8) P4 line 29: what is the purpose of scaling the timeseries by the grid cell areas?**

Author's reply:
*In the PCA the area of the grid cell should be taken into account to scale the time series as grid cell sizes depend on their latitudinal position, so that e.g. the grid cells at 20N have a larger weight compared to the ones at 50N.*

Changes in text:
No changes.

Comment of the referee:
**9) P6 lines 7-12: this list of other methods is interesting, but makes the reader wonder why the authors chose the anylitic signal method instead of these. Could you please clarify this choice?**

Author's reply:
*P6 lines 7-10 lists alternatives for the phase synchronization analysis as it is done here using the phase difference evolution and a statistical test based on surrogates. We chose this way of investigating phase synchronization in the Kuroshio Current system because the plateaus in the phase difference evolution as epochs of synchronization are very intuitive compared to the measure of*

*Shannon entropy (Palus et al., 2007), the mean resultant path length (Palus et al., 2007), the test proposed by Vejmelka et al. (2007) and the mutual information (Jaycay et al., 2018, Palus et al., 2014 a,b). In addition, it has already been applied in other geophysical problems before (Gelbrecht et al., 2018; Maraun & Kurths, 2005; Palus & Novotna, 2006).*
*P6 lines 10-12 focuses on the possibilities of deriving the instantaneous phase from a one-dimensional time series which is needed to perform the phase synchronization analysis (or other methods mentioned in P6 7-10). The instantaneous phase can either be derived using the analytical signal approach (in combination with a SSA to do the following phase difference analysis for components of the time series with a specific oscillatory component) or the wavelet transform (which directly gives (complex) coefficients related to a certain frequency, compare Palus & Novotna, 2006).*

*When the proposed method of phase synchronization analysis was performed previously, the instantaneous phase was mostly derived using the analytic signal approach (Gelbrecht et al., 2018; Maraun & Kurths, 2005; Felix et al., 2010), often in combination with a preceding SSA, instead of the wavelet transform so that we decided to follow this approach as well. Palus & Novotna (2006) perform their analysis for both the analytic signal method (in combination with a SSA) and the wavelet transform and found that both methods result in the same plateaus in the phase difference evolution in terms of their localization in time. However, the estimate of the phase difference evolution obtained by the wavelet transform was smoothed.*

Changes in text:
A sentence justifying the choice of the phase synchronization method will be added.

Comment of the referee:
**10) P7 line 5: how representative is a mode accounting for 29.6% of the explained variance? Is it comparable to other studies?**

Author's reply:
*This is actually quite larger (larger than in observational analyses) which is due to the relatively long data set from the CESM.*

Changes in text:
A remark on this will be made, referring to a typical observational study.

Comment of the referee:
**11) P7 lines 16-30. This part is difficult to follow for the non-specialist (that I am). I would be useful to give more explanations, about e.g. the nature and expected impact of M, the nature of the "ST-PCs", etc. Also, Figure 3's caption mentions "the first PC" and "a specific PC": are these different quantities? Please clarify these points.**

Author's reply:
*A singular spectrum analysis is done to extract certain significant oscillatory components from the considered time series for which then the phase synchronization analysis is applied.*
*In a singular spectrum analysis, the time series x(t) is shifted up to a predefined lag M, i.e. the time series is delay-embedded resulting in a MxN matrix where the kth row represents a time series with lag k. For the time-delay embedded time series the lag-covariance matrix is calculated and a PCA is performed for the lag-covariance matrix. The eigenvectors give the ST-EOFs and projections onto them result in the ST-PCs. To assess the significance of the St-PCs, they are tested against red noise surrogates in a Monte Carlo SSA. In addition, parts of the original time series can be reconstructed and a reconstructed component (RC) is obtained. For more (mathematical) details we refer to Ghil et al. (2002). The choice of M is a trade-off between applying a wide window to obtain as much information as possible and a window which includes many replicates of the time series properties of interest to assure statistical confidence. This trade-off is also discussed in Ghil et al. (2002).*

*We agree that the caption of Fig. 3 can be improved. The first PC refers to the PC obtained in the Principal Component Analysis of the zonal wind stress field by projecting the spatial EOF pattern onto the original anomaly field. The first PC of the zonal wind stress is used as a basis for representing the wind as one part involved in the synchronization. Fig. 3b shows the frequency spectrum of the first PC of the zonal wind stress. Instead of applying the synchronization directly to the first PC of the zonal wind stress, a Singular Spectrum Analysis is applied to extract certain oscillatory components of the zonal wind stress PC. Within the Singular Spectrum Analysis of the first PC of the zonal wind stress again (pairs of) ST-PCs are obtained which are displayed in Fig. 3d. Each (pair of) ST-PCs obtained in the Singular Spectrum Analysis is associated with a certain frequency.*

Changes in text:
We will add further explanations of the Singular Spectrum Analysis and its parameters in the text.

The caption of Fig. 3 will be changed to make the difference between PCs obtained in the simple PCA and the PCs resulting from the subsequent singular spectrum analysis clear. The text will be changed to "(a & b): Fourier spectrum (black curve) of the KE path length and the first PC of the zonal wind stress (obtained from the projection of the spatial EOF pattern). … (c & d): Data basis from the Monte Carlo SSA for the KE path length and the first PC of the zonal wind stress (obtained from the projection of the spatial EOF pattern). (e & f): Noise basis from the Monte Carlo SSA for the KE path length and the first PC of the zonal wind stress (obtained from the projection of the spatial EOF pattern). … A specific ST-PC resulting from the SSA is represented by … The ST-PCs related to the 8-year period are significant…".

Comment of the referee:
**12) P10 line 1: suggestion if more accurate: ": : :oscillates several times"**

Author's reply:
*Agreed.*

Changes in text:
Text will be changed to "The trajectory in the plane spanned by the time derivative of the observable and its Hilbert transform oscillates  several times around the origin so that…"

Comment of the referee:
**13) P10 line 4: how and why are these 3 values of M chosen? What does the detection of synchronisation for these specific 3 lag-window lengths imply physically (in terms of time-scales or lags, for instance)? What is happening for other M values?**

Author's reply:
*The best results in representing the 8-years period were obtained for M = 365. In general, SSA was applied for a whole range of values of M to check whether the results from the phase synchronization analysis depend on the choice of M in the previous SSA. Results are displayed in Fig. 4d showing the significance of the phase difference for a range of M-values while Fig. 4c exemplarily displays the phase difference evolution for 3 values of M.*
*Phase synchronization does not only occur for the values of M whose phase difference evolution is explicitly shown (in Fig. 4c) but also for other values of M (Fig. 4d). Statistically significant plateaus in the phase difference evolution are displayed in blue. Values of M outside the range which is considered here are not expected to give meaningful results with respect to the trade-off described in Ghil et al. (2002) and mentioned in the reply on comment **11).***

Changes in text:
No changes as this was already in the text. The explanation of the different phase differences with M will be dealt with in the next comment.

Comment of the referee:
**14) P10 line 4: could you please give an interpretation of the phase differences reached on the 3 "plateaus" (-0.75 and 0.25)? why do they differ among the 3 values of M, and what do these differences mean? How small should oscillations be around a given phase difference for an evolution to be labelled as a "plateau"?**

Author's reply:
*The actual value of the phase difference is less important than the fact that is constant over some time interval. The different values of the phase difference arise because different ST-PC arise for different M, with a slightly different period.*

*We totally agree that it is subjective to select the plateaus and to decide which oscillations are small enough for the phase difference evolution to be labelled as a „plateau". Hence, to avoid a selection of plateaus in the phase difference evolution by visual inspection and to obtain a quantitative evidence of phase synchronization in distinct epochs, we applied the moving window approach and the statistical test described in Gelbrecht et al. (2018). The distribution of the mapped back phase difference should be significantly different to a uniform distribution of phase differences for non-synchronized time series. The Kolmogrovo-Smirnov test was used to compare the phase difference distribution and for a low p-value of $p < 0.0001$ the null hypothesis was rejected.*

Changes in text: The text will be adapted to address the first issue raised here. The second issue was addressed in the text.

Comment of the referee:
**15) P10 line 9: it would be interesting to show the surrogate and actual "distributions" during synchronisation.**

Author's reply:
*The surrogates act as a null hypothesis of no synchronization between the wind stress in the North Pacific Basin and the KE path length. Between time series which are not synchronized no dominant phase difference occurs which results in a uniform distribution of the mapped-back phase differences. In contrast, phase synchronized time series show a preferred phase difference and a corresponding peak in the distribution of the mapped-back phase differences. As such a phase difference distribution for both considered time series is calculated for every sliding window applied to the time series, many histograms are obtained (depending on the sliding window length and the length of the time series). In addition, a series of histograms is obtained for every lag-window length M. Important information about the preferred phase difference is given with the phase difference evolution.*

Changes in text:
A remark on this will be made in the revised text.

Comment of the referee:
**16) P10 lines 13-18: Could you please clarify why finding dominant periods which are similar in both Fourier spectra limits the "risk" to spuriously detect a synchronisation? Line 16 indeed suggests that finding similar frequencies in both time series might expose to this risk.**

Author's reply:
*We agree that the point regarding the risk of similar dominant periods in the considered time series was not stated clearly. Plateaus in the phase difference evolution can be due to the process of synchronization. However, a 'spurious' constant phase difference evolution in time also occurs if the phase difference is calculated between time series which have the same period. Processing the time series with SSA allows to perform the phase synchronization analysis using a component of the original time series with a specific frequency (i.e. the frequency corresponding to the process which is analyzed).*

*If the time series obtained from SSA used in the phase synchronization analysis would have an identical frequency, such spurious plateaus in the phase difference evolution could occur. However, the spectra of the used time series for the phase synchronization analysis (i.e. the reconstructed components) show differences which limits the risk of detecting such spurious synchronization.*

Changes in text:
We will clarify this issue (risk of using SSA processed time series in the phase synchronization analysis) by reformulating the statement on the Fourier spectra of the time series.

Comment of the referee:
**17) * P10 lines 21-26: what is the implication of not finding synchronisation between the NPGO index and the KE path length? Doesn't it mean that the atmospheric pacing of intrinsic KCS variability is questionable in a way?**

Author's reply:
To us it implies that the NPGO pattern does not contribute much (compared to the PDO) to the excitation of Rossby waves which are responsible for the synchronization. Note that the NPGO is the second EOF (the PDO is the first EOF) and explains a smaller part of the variability.

Changes in text: A remark on this will be made in the revised text with support of further analysis that will be done on the CESM results.

Comment of the referee:
**18) P12 lines 16-17: suggestion: "with positive (negative) IPD anomalies, as expected from the baroclinic adjustment process."**

Author's reply:
*Agreed.*

Changes in text:
The text will be changed to "The Hovmöller diagrams also indicate that positive (negative) SSH anomalies coincide with positive (negative) IPD anomalies, as expected from the baroclinic adjustment process."

Comment of the referee:
**19) * P13 line 1: I do not understand the suggested dynamics, please clarify: through which simple process can the "stretching of the layer" "lead to" horizontal velocity fluctuations? Which component of the velocity? Is this a potential vorticity argument (which may lead to meridional velocity anomalies)? Alternately or in addition, would it be useful to try to relate the meridional gradient of SSH or IPD (which may lead to changes in upper zonal velocities, i.e. the strength of the zonal jet)? Trying to relate dynamicallyconnected fields could help interpret the link between fluctuations in SSH or IPD, the jet intensity (and perhaps the EKE).**

Author's reply:
*To clarify the connection of the isopycnal slope changes to the velocity anomalies in the proposed physical mechanism of the phase synchronization we carried out further analysis of the CESM output.*
*Fig. 4a&b show the zonal average of the zonal velocity with depth and depending on the latitude for the model years 174 – 176 and 181 – 183 (i.e. a three-year average). The zonal average was taken between 140°E and 160°E which is the region where the Kuroshio jet is located. In addition, the isopycnals are presented. A clear difference between the zonal velocity and the isopycnals can be observed for model years 174-176 and 181-183. The zonal velocity for model year 174 – 176 with a strong meandering state of the Kuroshio Current is much smaller compared to model year 181 – 183. The high zonal velocity for model year 181 – 183 is related to a straight state of the Kuroshio Current. In addition, the zonal velocity is very localized as a jet for model year 181 – 183 while it is*

*spread out along a wide range of latitudes for model years 174 – 176 with a meandering Kuroshio Current.*

*For model year 174 – 176 and model year 181 – 183 the isopycnals have a positive slope towards the north with relative warm water south of the Kuroshio Current and relative cold water north of the Kuroshio Current. The slope for model year 181 – 183 is, especially in the Kuroshio jet region (33-35°N), much larger than for model year 174 – 176 (related to the thermal wind balance).*

*To summarize, a strong slope of the isopycnals occurs with very localized high zonal velocities and a straight state of the Kuroshio in CESM. In contrast, low zonal velocities which are spread out along a wide latitudinal range, are related to a meandering Kuroshio Current and a less sloped isopycnals. This can be expected from the thermal wind balance.*

*We also calculated the evolution of the 1028 kg/m³ isopycnal in time (Fig. 4c) and a decadal variability of the isopycnal depth corresponding to the previously presented results (Fig. 6a in the original manuscript) can be observed.*

*Fig. 4d shows the maximum slope of the 1028 kg/m³ isopycnal in time (red curve). For the latitude at which the maximum slope of the 1028 kg/m³ isopycnal is reached the maximum zonal velocity with depth was determined (Fig. 4d, black curve). The time series of the maximum slope of the 1028 kg/m³ isopycnal and the maximum zonal velocity with depth are correlated well with a high correlation coefficient of 0.96.*

[Figure]

*Figure 4: (a & b): Zonal averaged (in region 140-160°E) zonal velocity with depth as well as isopycnal for model year 174-176 and model year 181-183 (3-year average). The thick black line is the 1028 kg/m³ isopycnal, the spacing between the isopycnals is 1 kg/m³. (c): Hovmöller diagram for the evolution of the 1028 kg/m³ isopycnal. (d): Time series of the maximum slope of the 1028 kg/m³ isopycnal and the maximum velocity at the maximum slope latitude.*

Changes in text:
We will adapt Section 3.3. regarding the proposed physical mechanism of phase synchronization, add additional results from the CESM analyses, and give a more detailed explanation on the link between the deepening of the IPD and the decreased velocity field based.

Comment of the referee:
**20) P13 line 13: if "it is not entirely clear which time series leads or lags", is it OK to state a few lines above that IPD leads EKE which leads the KE length? This is a bit confusing.**

Author's reply:
*Agreed.*

Changes in text:
The sentence will be adapted, as the lead/lag relation is clear from the physical processes.

Comment of the referee:
**21) * Last paragraph in section 3: this part is rather allusive and not solidly grounded physically (although I acknowledge that answering the following questions may not be easy given the chosen indices; see remark 19 above). - Through which process, or according to which dynamical balance, does the "deepening of the IPD [: : :] leads to a weaker flow" (line 14)? - Why only barotropic instability is mentioned, although isopycnal depth modulations more likely reveal changes in the 1st-mode baroclinic flow (and potentially in vertical shear and baroclinic instability strength)? It could be interesting to look at the meridional slope of the isopycnal, which relates physically to the zonal flow and to the available potential energy reservoir (hence potentially to EKE changes through baroclinic instability). - In line 16: what does "optimal" lag-correlation mean, and why is it expected to occur "at about 1/2 of the mean oscillation period"? - Finally (line 1 next page), it is not clear to me what "This view" includes precisely, given the large number of suggested links mentioned and the numerous statements made in this paragraph. Please clarify these various issues, and the dynamical arguments.**

Author's reply:
*See reply to comment 19).*

Changes in text:
See reply to comment 19). We will explain "this view" more thoroughly using the additional results from the CESM analyses.

Comment of the referee:
**22) P14 lines 19-20: "too large/small" with respect to what?**

Author's reply:
*This was meant in comparison to the intrinsic variability, but should indeed be clarified.*

Changes in text:
We will reformulate the sentence.

Comment of the referee:
**23) P14 line 20: I think that in this context ("strong" forcing) the KCS variability would be "periodic" only if the forcing is periodic as well. If yes, please clarify.**

Author's reply:
Correct.

Changes in text:
This will be mentioned, also in response to comment 22).

Comment of the referee:
**24) P14 line 21: "during a few intervals": only one interval of synchronisation is presented in the paper. When to the other intervals occur?**

Author's reply:
*For the case shown in Fig. 3c, the main interval showing phase synchronization is between model year 200 and 240. Results on the statistically significance of plateaus in the phase difference evolution in Fig. 3d indicate that some lag values M (mostly for $M \gtrsim 350$) also show a plateau in the phase difference evolution for a bigger time span of model year 190 to 290. In general, all time intervals which are not colored red have a statistically significant plateau in the phase difference evolution and Fig. 3d reveals several (smaller) intervals in addition to the main interval. However, the phase synchronization is, for the interval on which we mainly focused, very robust across the M values. Plateaus in the phase difference evolution during other intervals are not that robust across the M values so that the existence of phase synchronization cannot be clearly demonstrated.*

Changes in text:
The sentence will be reformulated accordingly.

Comment of the referee:
**25) P14 lines 22-23: I do not see a drop in the IPD index in years 240-250 (it seems to remain at about -10 on the plot, i.e. a rather moderate negative value compared to the rest of the time series). Also, are such IPD anomalies restricted to the coupled simulation (with interannual atmospheric variability), or are comparable anomalies also present in the climatological simulation (with no interannual forcing)? If comparable IPD changes occur in the climatological simulation, then one may not conclude that "the epochs of the phase synchronisation" are set by the "variation in the forcing", but by IPD changes, whether they be atmospherically driven or spontaneously produced**

Author's reply:
This needs further analysis, in particular for the climatological forced model.

Changes in text:
We will carry out this analysis and adapt the description accordingly.

---

## Author Comment (AC2) · 28 Nov 2019

**Response to Anonymous Referee RC 2**

We thank the referee for the careful reading and the useful comments and will adapt the manuscript accordingly. Below is a point by point reply with the referee's comments in bold font, our reply in italic font and the changes in manuscript in normal font.

Comment of the referee:
**MAJOR COMMENT: The open ocean generation of baroclinic Rossby waves requires two fundamental elements: time-dependence of the atmospheric forcing and perturbations of the ocean thermocline. The latter are produced by the vertical Ekman pumping associated with the horizontal divergence of the Ekman transport, which, in turn, is proportional to the vertical component of the curl of the surface wind stress So I wonder why the authors have chosen the zonal wind stress tau1 instead of curl(tau) as the representative observable for the at­mosphere. Their choice is even more surprising because, to investigate the synchronization they analysed "the vertical motion of the isopycnals" induced by Rossby waves in the KE region; this is absolutely correct but, then, why ignoring the same effect in the open ocean where the waves are generated? In my opinion the best thing would be to redo all the calcu­lations starting from curl(tau) instead of tau1. Alternatively, I recommend to calculate the first PC of the vertical component of curl(tau), compare it with Fig. 2d and show that the two time series are, in their turn, fairly well synchronized. This is possible; in fact, if on the one hand, in the extreme -unrealistic- case of latitude-independent zonal wind stress no wave genera­tion would occur, on the other hand, some degree of synchronization between the first PC of tau1 and that of curl(tau) can be expected. In any case I strongly recommend the authors to address this important issue.**

Author's reply:
*We have redone the phase synchronization analysis using the wind stress curl. In particular, we calculated the first PC of the wind stress curl in a PCA which is shown in Fig. 1d. After smoothing, it is similar to the one obtained for the zonal wind stress. The EOF pattern of the wind stress curl (Fig. 1b) is different to the one obtained for the zonal wind stress and the variance for the first EOF of the wind stress curl (7.6%) is much smaller compared to the zonal wind stress (29.6%). The spectrum of the first PC of the wind stress curl $\nabla \times \tau_M(t)$ (Fig. 2b) still shows a dominant period of 8 years. Afterwards, a SSA was applied to the first PC of the wind stress curl. The same dominant period of 8 years was found. The corresponding ST-PCs pair consist of ST-PC 3 and 4 (for the zonal wind stress the ST-PC pair associated with a period of 8 years this was 1-2) and is indicated in Fig. 2d & f. For the ST-PCs pair 3-4 the reconstructed component $\nabla \times \tau_M(t)$ was calculated. Note that the results of the SSA are given for a lag of M = 325 months in the case of the wind stress curl while for the zonal wind stress results were given for a lag of M = 375 months.*
*As a next step, a two-dimensional embedding of $\nabla \times \tau_M(t)$ with its Hilbert transform was done. Os­cillations of the trajectory in the plane spanned by the (time derivative) of $\nabla \times \tau_M(t)$ and its Hilbert transform around the origin can be observed (Fig. 3b).*
*Results for the phase synchronization analysis are shown in Fig. 3c & d for the $\nabla \times \tau_M(t)$ and the $L_M(t)$ (i.e. the time series obtained after performing a SSA for the KE path length). Fig. 3c shows the phase difference evolution between $L_M(t)$ and $\nabla \times \tau_M(t)$ for three lag-window lengths (M = 275, 325 and 365 months). As for the zonal wind stress, a (significant) plateau can be identified between model year 200 and 240 for all three values of M indicating a phase synchronization. Epochs with plateaus in the phase difference evolution are confirmed by the statistical test using a sliding window of 120 months length (Fig. 3d). The phase synchronization between model year 200 and 240 occurs for various values of lags M. For lags M > 340 Fig. 3d also reveals a plateau for model years 250 to 290.*
*All in all, using the wind stress curl in the phase synchronization analysis resulted in very similar results as for the zonal wind stress with some minor changes such as the precise ST-PCs pair.*

[Figure]

a) Sea surface height, model year 250.

b) EOF 1, $\nabla \times \tau$

c) KE path length

d) PC 1, $\nabla \times \tau$

Figure 1: Sea surface height field in the North Pacific basin for January model year 250. The black curve is the 70 cm SSH isoline that represents the KE jet. (b): First EOF of the wind stress curl over the North Pacific basin, explaining 7.6 % of the total variance. (c): Time series of the KE path length along the region 140°E – 60°E, together with its moving average of 36 months (red curve). (d): First PC of the wind stress curl over the North Pacific basin, together with its moving average of 36 months (red curve).

[Figure]

Figure 2: (a & b): Fourier spectrum (black curve) of the KE path length and the first PC of the wind stress curl. The colour-coded curves are the significance levels of red noise surrogates, where the number (in years) indicate significant periods. (c & d): Data basis from the Monte Carlo SSA for the KE path length and the first PC of the wind stress curl. (e & f): Noise basis from the Monte Carlo SSA for the KE path length and the first PC of the wind stress curl. SSA was performed with a total lag of 325 months and for the Monte Carlo SSA 2500 realizations have been used. The 95% confidence intervals which have been calculated from the surrogates are shown by the red vertical bars. A specific ST-PC is represented by the blue markers and is associated with a certain frequency. The ST-PCs related to the 8-year period are significant in both the data basis and the noise basis (Allen & Robertson, 1996) and the ST-PC pair related to this variability is indicated.

[Figure]

Figure 3: (a & b): Embedding of the time derivative of the KE path length time series $L_M(t)$ (a) and that of the first PC of the wind stress curl (b) by the Hilbert transformation. (c): Phase difference between $L_M(t)$ and $\nabla \times \tau_M(t)$ for three lag-window lengths M. The black curves indicate starting points of the sliding window (length of 120 months) with p-value < 0.0001. Intervals that are not significant (p-value ≥ 0.0001) are indicated in red. The black horizontal lines indicate the values of M used in (c).

Changes in text:
A remark on the similarity of the phase synchronization analysis for the zonal wind stress and the wind stress curl will be added.

MINOR COMMENTS

Comment of the referee:
**The mathematical method used in the analysis is relatively complex and requires the implementation of several steps, so it is not easy to follow the description. Besides, the listing of the technical steps does not end in sect. 2.2. but continues in sect. 3.1 on the variability. In particular, it is difficult to follow lines 8-30 of page 7. Therefore I suggest to compact the discussion of all the technical details in sect. 2.2 and to add a flowchart summarizing the mathematical technique. This would be very helpful, especially because I believe that the Hilbert method does not belong to the typical background of oceanographers.**

Author's reply:
*Section 2.2. aims to give a general introduction to the main method used to study phase synchronization effects including the Hilbert transform to derive the phase of the time series, the calculation of the phase difference as well as the statistical test to determine (statistically significant) epochs of phase synchronization. In Section 3.1. information on the processing (detrending, removing the seasonal cycle, and SSA) of the time series chosen as representatives for the proposed synchronizing processes (i.e. for which the phase synchronization analysis introduced in Section 2.2. will be done) is given and their variability is described. We agree that it might be confusing that the information on the processing of the time series is given after the introduction of the phase synchronization method (which comes later in the work flow).*

Changes in text:
We will reorganize the description of the technical steps given in Section 2.2. and Section 3.1. to clarify the methods used to study the Kuroshio Current system.
In addition, we will add further explanations of the Singular Spectrum Analysis and its parameters in the text.

Comment of the referee:
**p. 1, l. 22-24: the description of the elongated KE mode corresponds to that of the contracted mode, and vice versa, please exchange.**

Author's reply:
*Agreed.*

Changes in text:
The text will be changed to "An  contracted ( elongated) state is characterized by...".

Comment of the referee:
**p. 2, l. 15-16: I do not think a critical effect of topography has been recognized to determine "An intensified mesoscale eddy field". Please provide references.**

Author's reply:
*Qiu & Chen (2010) describe that a southward shift of the KE jet results in interactions with the shallow topography of the Shatsky Rise which eventually leads to an increase of the eddy kinetic energy state of the KE system west of the Shatsky Rise. As a consequence of the intensified mesoscale eddy field, the southern recirculation gyre is strengthened.*

Changes in text:
We will provide the corresponding references for this statement.

Comment of the referee:
**p. 3, l. 9-11: I do not fully agree with the sentence: ": : : focussing on what we think is a missing piece to reconcile the forced and internal views as sketched above: a phase synchronisation of the KCS variability with the zonal wind-stress variability in the North Pacific". Such phase synchronization has indeed been investigated in several modelling studies, several of them also quoted by the authors. I would therefore suggest to use a more moderate sentence, and similarly elsewhere.**

Author's reply:
*We agree that modelling studies with the aim to reconcile the forced and the internal view on the KCS variability exist and also provide an investigation of the relation between the wind stress and KCS variability. On the other hand, an explicit framing of the KCS variability in terms of phase synchronization as it is described in section 5.2 of Pikovsky et al. (2001) and especially using the corresponding method from nonlinear dynamics (phase difference evolution and the statistical test) to address the phase synchronization cannot be found. In addition, in the meantime we were pointed to work by Andrew Kiss (Kiss and Frankcombe, 2016, J. Clim) where they study synchronization mechanisms (between time dependent wind stress and western boundary current response) in idealized QG models.*

Changes in text:
We will adjust the statement (and similar statements elsewhere) to account for the existing modelling studies and clarify the difference to our contribution by the phase difference approach used. We will also include the work using idealized models.

Comment of the referee:
**p. 4, l. 8: define the path length parameter in the text (now it is defined only in the caption to Fig. 1) and discuss why it has been chosen among the many available parameters applied to the KE variability in previous studies.**

Author's reply:
*The KE path length is derived from the 70 cm SSH isoline along the region 140°E – 160°E. The respective isoline can be identified in the sea surface height field in the North Pacific basin displayed in Fig.1. The path length was shown to be a very good indicator of the different KCS states in many previous studies (Qiu & Chen, 2010, Pierini et al., 2009) and is therefore chosen.*

Changes in text:
The definition of the KE path length given in the caption of Fig.1 is indirectly given in the text (p. 4, l. 7). We will change the respective sentence to "In CESM, the KE path length is based on the 70-cm SSH contour in the region along 140°E – 160°E". We will more clearly motivate the use of the KE path length as a parameter for the KE variability on p. 4 (where the two time series representing the processes which are supposed to synchronize are introduced).

Comment of the referee:
**p. 12, l. 12-14: "The travel time of these Rossby waves from the eastern to the western side of the North Pacific basin is estimated to be roughly 10 years corresponding to the observed decadal variability of the Kuroshio oscillator." First of all the travel time of 10 years is much large than that observed by several authors (which is of 2-3 years). Secondly, the decadal variability of the KCS may be linked to that of the atmosphere but not to the travel time of the waves. Please clarify the first point and reformulate the sentence.**

Author's reply:
*We do not agree. In the original Chelton and Schlax (Science 2008) paper, typical Rossby wave travel times at 30 N are about 10 years.*

Changes in text:
The reference to the Chelton and Schlax 2008 paper will be included and a remark on the time scale will be made.

Comment of the referee:
**Section 4 includes, summary, discussion and conclusions. I think it would be much better to move the discussion about some crucial points (e.g., p. 14, l. 19-24, 28-31) in section 3.3.**

Author's reply:
*Section 4 is structured as following: The first paragraph summarizes the goal and main results of the study. Then the phase synchronization and its coupling / forcing mechanism is addressed. In this context, it is discussed why the processes are not synchronized during the whole model time and an explanation is proposed (related to the strength of the forcing). This could, of course, also be discussed in the section 3.2. The next paragraph focuses on the Rossby waves and give climatologically forced model results as a supporting argument for the mechanism of synchronization. Finally, an outlook is given on the demonstration of the synchronization phenomenon using real data.*

Changes in text:
As further explanations / clarifications related to p. 14 l. 19-24 will be added to Section 4 (Comments of Referee 1) we will consider whether it will be appropriate to reorganize parts of Section 4.

Comment of the referee:
**p. 13, l. 16: his -> is**

Author's reply:
*Agreed.*

Changes in text:
Text will be changed to "The optimal-lag correlation  is expected to occur at about 1/2 of the mean oscillation period, which is about 4 years, consistent with Figure 6b."

---

## Referee Report (RR1)

In my major comment I recommended to either redo all the calculations starting from curl(tau) instead of tau1 or, alternatively, to calculate the first PC of the vertical component of curl(tau), compare it with Fig. 2d and show that the two time series are, in their turn, fairly well synchronized. I am glad to see that the authors have followed the first, more demanding but more appropriate alternative. It is also interesting to see that similar results are obtained, as it could in fact be expected.

In consideration of both the authors' response and the revised manuscript, I can say that all the minor points I raised have been appropriately addressed. I would just like to point out that in my comment on the travel time of Rossby waves, which I indicated as being 2-3 years, I had in mind the central North Pacific (e.g., see the Hovmoller diagram in Fig. 8b of Qiu and Chen, 2005) rather than the eastern side of the ocean, for which 10 years is clearly the correct travel time estimate.

In conclusion, in my opinion the revised manuscript can be accepted for publication in its present form.

---

## Author Response (AR2)

**Response to Anonymous Referee RC 1**

We thank the referee for the additional useful comments and have adapted the manuscript accordingly. Below is a point by point reply with the referee's comments in bold font, our reply in italic font and the changes in manuscript in normal font.

Comment of the referee:
1) **section 3.3 is now clearer: Figure 6 confirms that zonal velocity and meridional isopycnal slopes covary strongly within the jet, although it was expected from geostrophy. However, this section remains unprecise, in particular about the nature and role of the "rectification" process, mentioned in page 16 (lines 1 and 3). For instance, the expression "rectification effects due to barotropic instabilities" is a bit confusing and sounds contradictory, since rectification (according to e.g. Qui and Chen, 2010) induces a EKE—to-MKE flux while barotropic instability indices the opposite flux. Could you please clarify the role and nature of "rectification" in the suggested dynamical scenario?**

**In addition, I had asked in my previous review (comment #21) "Why only barotropic instability is mentioned?" in this section, but it turns out that baroclinic instability is still not mentioned there (despite its important role in producing mesoscale eddies whose EKE, in turn, are then thought to rectify the jet).**

**In summary, a more careful explanation of the processes that control [1] the Kuroshio intrinsic variability, and [2] its modulation by the atmosphere (which is the subject of the paper) would clearly improve the paper. It would be very useful in this exercise to consolidate the evidence that supports these statements, and if the suggested dynamical scenario(s) differ from the published literature.**

Author's reply:
*Yes, we agree that baroclinic instability is also important in the energy conversion.*

Changes in text:
We have adapted and slightly extended the synchronisation mechanism in section 3.3.

Comment of the referee:
2) **About my former comment #25 about the "drop in IPD index in years 240-250". The authors have not responded to my first sentence, where I said that I cannot see any drop (although I may not look at the right plot?). Please make sure that the drop occurs, clarify when and in which figure it can be seen, and try to reformulate the subsequent argument in the paper.**

Author's reply:
*We did not respond to this because this earlier description was not correct and hence deleted in the text of the revised paper. We forgot to mention this in the reply.*

Changes in text:
No changes in the text.

Comment of the referee:
3) **Page 3 line 10 : ...compared to the climatological simulation.**

Author's reply:
*Agreed.*

Changes in text:
Text is changed accordingly.

Comment of the referee:
**4) page 10, legend fig 3 : ...obtained from the projection.**

Author's reply:
*Was indeed a bit unclear.*

Changes in text:
Text is changed accordingly.

Comment of the referee:
**5) page 12 line 13 : ...which is considerably smaller than the variance explained by the first PC...**

Author's reply:
*Agreed. Thanks for the suggestion.*

Changes in text:
Text is changed accordingly.

Comment of the referee:
**6) end of page 12 and beginning of page 13 : shouldn't it read : "...detected from observations." ?**

Author's reply:
*Indeed, this is better.*

Changes in text:
Text is changed accordingly.

Comment of the referee:
**7) page 14 line 5 : ... zonal velocity and density ...**

Author's reply:
*Suggestion followed.*

Changes in text:
The word 'water' is deleted.

Comment of the referee:
**8) page 14 lines 7-8 : it seems to me that years 174-176 and 181-183 correspond to contracted and elongated states, respectively. Am I right ?**

Author's reply:
*Yes, thanks; these got mixed up.*

Changes in text:
Text is corrected.

Comment of the referee:
**9) Page 14 line 9 : Relatively warm...**

Author's reply:
*OK.*

Changes in text: Text is changed accordingly.

Comment of the referee:
**10)** Page 14 line 30 : We interpret the results in Figures 6 and 7...

Author's reply:
*OK.*

Changes in text:
The word 'the' is deleted.

Comment of the referee:
**11)** About my former comment #10 about the 29.6% explained variance : the authors respond that a remark will be made "referring to an observational study", but I could not find anything about this in the revised version.

Author's reply:
*Yes, we forgot this in the revision.*

Changes in text:
We now mention the explained variance of the EOF (25%) in the observational study of the PDO in Deser et al. (2010), their Figure 10.

[revised manuscript text omitted]